# Model discovery to link neural activity to behavioral tasks

**Jamie D Costabile[1†‡], Kaarthik A Balakrishnan[1,2†], Sina Schwinn[1§], Martin Haesemeyer[1]***

[1]Department of Neuroscience, The Ohio State University College of Medicine, Columbus, United States; [2]Interdisciplinary Biophysics Graduate Program, Columbus, United States

**Abstract** Brains are not engineered solutions to a well-defined problem but arose through selective pressure acting on random variation. It is therefore unclear how well a model chosen by an experimenter can relate neural activity to experimental conditions. Here, we developed 'model identification of neural encoding (MINE).' MINE is an accessible framework using convolutional neural networks (CNNs) to discover and characterize a model that relates aspects of tasks to neural activity. Although flexible, CNNs are difficult to interpret. We use Taylor decomposition approaches to understand the discovered model and how it maps task features to activity. We apply MINE to a published cortical dataset as well as experiments designed to probe thermoregulatory circuits in zebrafish. Here, MINE allowed us to characterize neurons according to their receptive field and computational complexity, features that anatomically segregate in the brain. We also identified a new class of neurons that integrate thermosensory and behavioral information that eluded us previously when using traditional clustering and regression-based approaches.

***For correspondence:**
haesemeyer.1@osu.edu

[†]These authors contributed equally to this work

Present address: [‡]Hitachi Solutions America, Ltd, Irvine, United States; [§]Mass Eye and Ear, Massachusetts General Brigham, Boston, United States

## Editor's evaluation

This useful article describes a sensitive method for identifying the contributions of different behavioral and stimulus parameters to neural activity. The method has been convincingly validated using simulated data and applied to example state-of-the-art datasets from mouse and zebrafish. The method could be productively applied to a wide range of experiments in behavioral and systems neuroscience.

## Introduction

Contemporary neuroscience generates large datasets of neural activity in behaving animals (*Engert, 2014*; *Musall et al., 2019b*; *Urai et al., 2022*). To gain insight from these large-scale recordings, it is desirable to identify neurons with activity related to the behavioral task at hand. A common approach to this problem is to intuit a functional form ('a model') that relates predictors such as sensory stimuli, motor actions, and internal states to neural activity. Neurons can subsequently be classified into those with activity explained by features of the task and the chosen model and those with activity likely to be unrelated or background noise. A simple yet powerful approach is to use linear regression to explain neural activity as the weighted sum of sensory and motor features recorded during the experiment (*Miri et al., 2011*; *Harvey et al., 2012*; *Portugues et al., 2014*; *Musall et al., 2019a*). Since linear regression can accommodate nonlinear transformations of input variables, this technique can encompass diverse relationships between predictors and neural activity. Similarly, more flexible solutions using basis functions (*Poggio, 1990*; *Hastie et al., 2009*) can be used to identify task-related neurons. However, brains are not engineered solutions to a well-defined problem but arose through selective

pressure acting on random variation (*Eliasmith and Anderson, 2002*; *Niven and Laughlin, 2008*; *Zabihi et al., 2021*). It is therefore unclear how well neural activity can be captured by a well-defined function chosen by the experimenter as the test model.

Artificial neural networks (ANNs) can in principle accommodate any mapping of predictors to neural activity (*Hornik et al., 1989*; *Gorban and Wunsch, 1998*). At the same time, they can be designed to generalize well to untrained inputs (*Anders and John, 1991*; *Srivastava et al., 2014*) often over-coming problems related to explosion of variance and overfitting to training data associated with other solutions incorporating large numbers of parameters (*Hastie et al., 2009*; *James et al., 2013*). Due to this flexibility, insights into nonlinear receptive fields of visual and auditory neurons (*Lehky et al., 1992*; *Lau et al., 2002*; *Prenger et al., 2004*; *Ukita et al., 2019*; *Keshishian et al., 2020*) and into the encoding of limb motion in somatosensory cortex have been gained using ANNs (*Lucas et al., 2019*). However, an obvious drawback of ANNs is that they are much harder to interpret than models based on intuition and data exploration.

Here, we introduce 'model identification of neural encoding' (MINE). MINE combines convolutional neural networks (CNNs) to learn mappings from predictors (stimuli, behavioral actions, internal states) to neural activity (Figure 1) with a deep characterization of this relationship. This allows discovering a model or functional form from the data that relates predictors to activity and to subsequently describe this model, thereby inverting the usual approach. Using Taylor expansion approaches, MINE reveals the computational complexity such as the nonlinearity of the relationship (Figure 2), characterizes receptive fields as indicators of processing (Figure 3), and reveals on which specific predictors or their interactions the neural activity depends (Figure 4). By incorporating a convolutional layer, temporal or spatial transformations of inputs introduced by the technique (such as calcium indicator effects) or by the brain (such as differentiation of a signal, edge detection) will be learned seamlessly by MINE. These transformations therefore do not have to be captured through a priori transformations of the task variables. While the architecture and hyper-parameters of the CNN used by MINE impose limits on which relationships can be modeled, we consider the convolutional network largely 'model-free' because it does not make any explicit assumptions about the underlying probability distributions or functional forms of the data.

Here, we demonstrate the utility of MINE using a ground-truth dataset (Figures 1–4) and a cortical mouse widefield imaging dataset (Figure 5). We then designed a set of experiments to exhaustively probe thermoregulatory circuits in larval zebrafish (Figures 6 and 7). Specifically, we exploit the flexibility of MINE to provide randomly varied temperature stimuli across zebrafish and imaging planes while maintaining the ability to identify functional groups of neurons based on features of the trained CNNs. Using MINE, we discover a new functional class of neurons integrating thermosensory with behavioral information. Combining MINE with anatomical analysis, we also map functional features derived with MINE to a standard zebrafish brain.

## Results
### A model discovery approach to identify task-relevant neurons

MINE uses CNNs to overcome the challenges with predefined models while maintaining interpretability. Feed-forward ANNs are capable of approximating any function (*Cybenko, 1989*; *Hornik et al., 1989*; *Gorban and Wunsch, 1998*) and therefore afford great flexibility in capturing the relationship between 'predictors' (sensory stimuli, behavioral actions, internal states, etc.) and the activity of individual neurons. We designed a simple three-layered network architecture (*Figure 1A*). It consists of a linear convolutional layer (to accommodate temporal transformations such as calcium indicator effects) and two dense layers that capture nonlinear transformations. The architecture's simplicity speeds up training and eases interpretation while capturing transformations across time (i.e. the convolutional layers) and nonlinear effects (i.e. dense layers with nonlinear activations functions). We chose a continuously differentiable activation function for our dense network layers. Unlike the popular ReLu nonlinearity, this allows us to calculate higher-order derivatives that capture interaction effects and that allow us to quantify the computational complexity of transformations. Most network hyperparameters including the specific activation function ('swish,' *Ramachandran et al., 2017*) were determined by minimizing test error on a small dataset (see 'Methods'). We chose a fixed length of

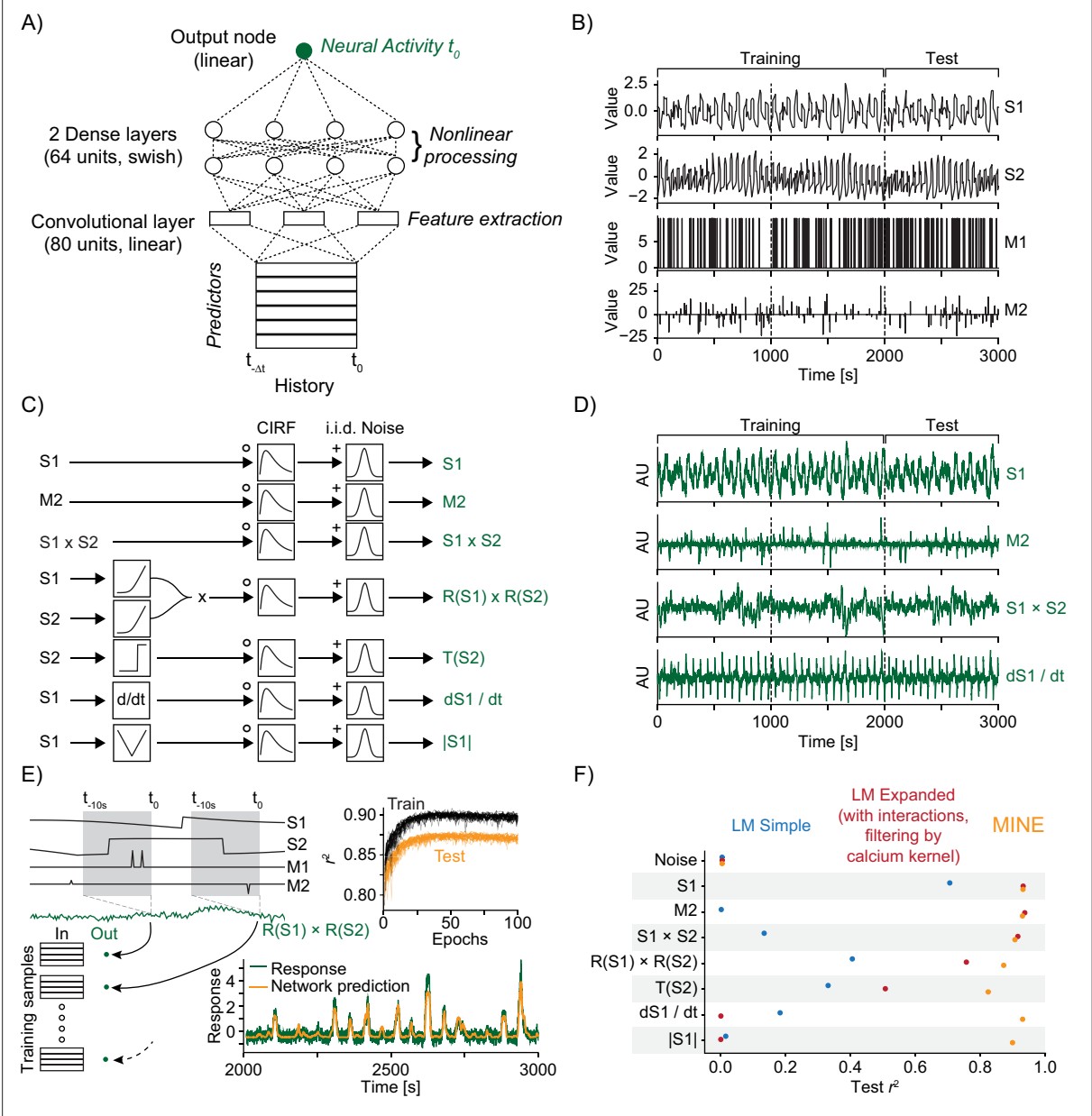

**Figure 1.** Model identification of neural encoding (MINE) accommodates a large set of predictor–activity relationships. (**A**) Schematic of the convolutional neural network (CNN) used. (**B**) The predictors that make up the ground-truth dataset. S1 and S2 are continuously variable predictors akin to sensory variables while M1 and M2 are discrete in time, akin to motor or decision variables. Dashed lines indicate a third each of the data with the first two-thirds used for training of models and the last third used for testing. (**C**) Schematic representation of ground-truth response generation. (**D**) Example responses in the ground-truth dataset. Labels on the right refer to the response types shown in (**C**). (**E**) The model predicts activity at time $t$ using predictors across time $t - \Delta t$ to $t$ as inputs. The schematic shows how this is related to the generation of training and test data. Top inset shows development of training and test error across training epochs for 20 models trained on the R(S1) × R(S2) response type. Bottom inset shows example prediction (orange) overlaid on response (dark green). (**F**) Squared correlation to test data for a simple linear regression model (blue), a linear regression model including first-order interaction terms and calcium kernel convolution (red), as well as the CNN fit by MINE (orange). Each dot represents the average across 20 models. While the standard deviation is represented by a dash, it is smaller than the dot size in all cases and therefore not visible in the graph.

The online version of this article includes the following figure supplement(s) for figure 1:

**Figure supplement 1.** Model identification of neural encoding (MINE) accommodates a large set of predictor–activity relationships.

10 s for our convolutional filters, which can be adjusted as needed, to capture temporal effects in all subsequent experiments.

To test MINE's ability to capture neural transformations and encoding of predictors, we generated a ground-truth dataset. This dataset consists of four randomly generated predictors (*Figure 1B*). Two of these, S1 and S2, vary continuously in time mimicking sensory stimuli. The other two, M1 and M2, are discrete like motor or decision variables. From these predictors, we generated 'neural responses' that depend on single predictors or interactions with and without intervening nonlinear transformations (*Figure 1C* and *Figure 1—figure supplement 1A*). We added Gaussian noise to all neural responses after convolution with a calcium kernel to approach conditions that might be observed in functional calcium imaging experiments (*Figure 1D*). For each neuron in our ground-truth dataset, we subsequently trained a CNN to predict the activity from the predictors (*Figure 1E*). To assess generalization, we split our dataset into a training set (two-thirds of the data) and a validation set (last third) (*Figure 1B and D*). Training for 100 epochs led to excellent predictions of 'neural activity' while maintaining generalizability as assessed by the squared correlation ($r^2$ value) to the test data (*Figure 1E and F*).

We sought to compare our ANN-based approach to another widespread approach to model single-neuron activity, linear regression. Using the four predictors (*Figure 1B*) as inputs, a simple linear regression model fails to explain activity in most cases (*Figure 1F* and *Figure 1—figure supplement 1C*). This is expected since the chosen type of linear regression model cannot learn the dynamics of the calcium indicator unlike the CNN. We therefore constructed an alternative model in which we convolved the predictors with the known 'calcium kernel' (*Figure 1—figure supplement 1B*). In this expanded linear model, we also included all first-order interaction terms by including pairwise products between predictors (*Figure 1—figure supplement 1B*). This type of model, capturing interactions and accounting for an estimated indicator effect, is popular in the analysis of large-scale calcium imaging datasets (*Miri et al., 2011*; *Ahrens et al., 2012*; *Portugues et al., 2014*; *Chen et al., 2018*; *Stringer et al., 2019*). As expected, this model matches the performance of the CNN in more response categories including nonlinear interactions (*Figure 1F*). However, the function of this model was designed using a posteriori knowledge about the responses. Nonetheless, the expanded linear model is poor in capturing some nonlinear transformations of predictors and fails to capture responses that relate to the time derivative of an input, for example, as expected in adapting neurons (*Figure 1F*). While other models could clearly be designed to overcome these challenges, this further illustrates the point that a model-based approach is limited to the constraints of the chosen model.

As shown, MINE can identify responses that depend on predictors independent of the linearity of these relationships. The underlying CNN is able to learn temporal transformations of inputs such as shifts in time or convolutions. Otherwise these transformations have to be explicitly provided to a regression model or the predictor matrix has to be augmented for the model to implicitly learn them (see the comparison model used in the analysis of zebrafish data below) (*Miri et al., 2011*; *Musall et al., 2019a*). MINE removes the requirement of estimating calcium response kernels that might differ across different neurons and can also identify responses that depend on derivatives of the input. This means that one predictor such as position can be used to identify neurons that depend on the velocity or acceleration of a stimulus, without the need of augmenting the predictor matrix.

## MINE characterizes computational complexity

Linear computations are limited in their expressivity, and it is generally believed that the computational power of the nervous system depends on nonlinear computation (*Hubel and Wiesel, 1968*; *Churchland et al., 1994*; *Carandini et al., 2005*). In spite of this, the responses of many neurons tend to depend almost linearly on sensory or behavioral features (*Miri et al., 2011*; *Thompson et al., 2016*; *Pho et al., 2018*; *Musall et al., 2019a*). The latter idea aligns with the hypothesis that information important to the animal should be linearly decodable by neural circuits (*Marder and Abbott, 1995*; *Eliasmith and Anderson, 2002*; *Shamir and Sompolinsky, 2006*). Disambiguating linear from nonlinear processing therefore provides important insight into circuit structure and function. Once fit to neural data, our CNNs model the transformations from predictors to neural activity (or from neural activity to action). We therefore set out to quantify the complexity of the function these networks implement as a proxy to classifying the actual transformations between stimuli, neural activity, and behavior.

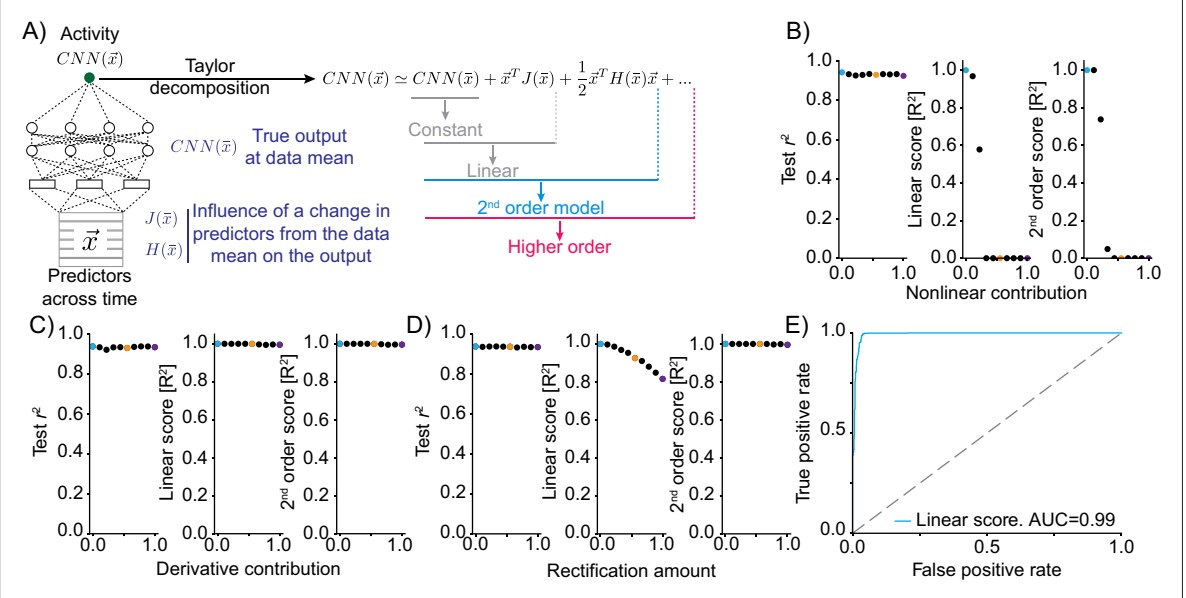

**Figure 2.** Truncations of the Taylor expansion assign computational complexity. (**A**) Schematic of the approach. At the data mean, the output of the network is differentiated with respect to the inputs. The first-order derivative (gradient J) and the second-order derivatives (Hessian H) are computed at this point. Comparing the output of truncations of the Taylor expansion can be used to assess the computational complexity of the function implemented by the convolutional neural network (CNN).For example, if the function is linear, it would be expected that a truncation after the linear term explains the vast majority of the variance in the true network output. (**B**) Mixing varying degrees of a nonlinear response function with a linear response ('Nonlinear contribution') and its effect on network performance (left, squared correlation to test data), the variance explained by truncation of the Taylor series after the linear term (middle) and the variance explained for a truncation after the second-order term (right). Colored dots relate to the plots of linear correlations in *Figure 2—figure supplement 1*. (**C**) As in (**B**) but mixing a linear function of a predictor with a linear transformation of the predictor, namely the first-order derivative. (**D**) As in (**B**) but mixing a linear function of a predictor with a rectified (nonlinear) version. (**E**) ROC plot, revealing the performance of a classifier of nonlinearity that is based on the variance explained by the truncation of the Taylor series after the linear term across 500 independent generations of linear/nonlinear mixing.

The online version of this article includes the following figure supplement(s) for figure 2:

**Figure supplement 1.** Truncations of the Taylor expansion assign computational complexity.

To arrive at a metric of 'computational complexity,' we used Taylor expansion to approximate the function implemented by the CNN. The Taylor expansion approximates a function, such as the CNN, around a specific input by a polynomial of up to infinite order. The weights of individual terms in this polynomial are determined by the values of the derivatives of the CNN output with respect to each input element (in our case predictors and timepoints). These derivatives describe the expected change in the output of the CNN given a small change in the predictor input. Calculating derivatives of increasing order at the point of expansion allows predicting the value of the CNN output at other points (akin to predicting the position of a car in the near future based on its current location, velocity, and acceleration). We specifically compute the first- and second-order partial derivatives of the CNN output with respect to each feature (predictor and timepoint) in the input at the average of the training data. This allows formulating the Taylor expansion of the network around the data mean. It is then possible to compare the quality of Taylor expansions with variable numbers of terms in predicting the true network output. If the network were to implement a linear function, the first-order term (the gradient $J$ of the function with respect to the input) should suffice to explain a large fraction of the variance in the output (*Figure 2A*). Nonlinear functions should depend on higher-order terms such as the second-order partial derivatives, $H$. We chose to define 'computational complexity' based on the requirement of the second- or higher-order terms in the Taylor expansion (*Figure 2A*). Specifically, we assign complexity 0 if the linear term in the expansion is enough to explain the activity, 1 if the quadratic term is needed, and 2 if higher-order terms are required. Nonlinear neurons are therefore split into two categories, depending on the complexity of the transformation.

We tested these metrics on ground-truth data in the following manner: we mixed predictors to varying degrees with either linear (as a control) or nonlinear transformations of the same predictors. By

increasing the contribution of the transformed predictor, we thereby inject varying amounts of nonlinearity into the response. We then applied MINE, training a CNN to learn the transformation between the predictor and the mixed output and calculating the coefficients of determination ($R^2$) for truncations of the Taylor series after the linear as well as the second-order term. Increasing the contribution of an arbitrary nonlinear function (i.e. $\tanh^2(x^3)$) leads to a marked decrease in both of these $R^2$ values (*Figure 2B*). Importantly, these metrics do not simply quantify a loss of linear correlation. Calculating the derivative of a function is a linear operation, and increasing the contribution of a derivative of the predictor indeed does not decrease either $R^2$ value in spite of reducing linear correlation between the predictor and the response (*Figure 2C* and *Figure 2—figure supplement 1A–C*). When increasing the contribution of a rectified linear version of the predictor, the amount of variance explained by the linear truncation drops while this is not the case for the truncation after the second-order term (*Figure 2D*). This is in contrast to the contribution of $\tanh^2(x^3)$ above and justifies our interpretation of computational complexity as a metric for how much a relationship diverges from linearity.

To quantify its usefulness to distinguish linear and nonlinear transformations, we systematically evaluated the classification performance of the fraction of variance explained by the linear truncation of the Taylor expansion. To this end, we generated random linear and nonlinear transformations of the predictors (see 'Methods'), trained the CNN, and calculated the $R^2$ value of the linear truncation. We compared this $R^2$ value to two other metrics of nonlinearity: the curvature of the function implemented by the network and the nonlinearity coefficient *Philipp and Carbonell, 2018*; *Philipp, 2021* of the network (see 'Methods'). Notably, a decrease in the explained variance by the linear truncation led on average to increases in both these metrics of nonlinearity (*Figure 2—figure supplement 1D and E*). To quantify classification performance, we used ROC analysis. This analysis revealed that the $R^2$ value ranks a nonlinear transformation lower than a linear one in 99% of cases (*Figure 2E*). Importantly, this metric allows disambiguating linear and nonlinear processing with acceptable false-positive and false-negative rates ($< 0.05$) at a cutoff of $R^2 < 0.8$ as a nonlinearity threshold *Figure 2—figure supplement 1F*; a feature that is highly desirable when applying the classification to real-world data.

In summary, MINE increases the interpretability of the CNN model and classifies the transformations the model encapsulates according to their computational complexity. Since the CNN encapsulates the transformations that are enacted by neural circuits between stimuli and neural activity or neural activity and behavior, this information provides important insight into the computations that give rise to neural activity.

## MINE characterizes neural receptive fields

Receptive fields compactly describe stimulus features that drive a neuron's response (*Dayan and Abbott, 2001*). They reveal integration times (those times over which coefficients of the receptive field are different from zero) and zero-crossings within the receptive field signal adapting responses that indicate that a neuron encodes the derivative of a stimulus across time (e.g. velocity encoding neurons) or that it detects edges across space. It is therefore desirable to use MINE to extract receptive fields of the neurons fit by the CNN. Because of their descriptive nature, different methods have been developed to extract receptive fields, commonly referred to as 'system identification' approaches. The Wiener/Volterra expansion of functions provides a powerful framework for system identification (*Poggio and Reichardt, 1973*; *Aertsen and Johannesma, 1981*; *Friston et al., 1998*; *Mammano, 1990*; *Marmarelis, 2004*; *Mitsis et al., 2007*; *Mitsis, 2011*). Since the input to the CNN at the heart of MINE contains information about each predictor across time, the Taylor expansion introduced in the previous section (*Figure 2A*) is equivalent to the Volterra expansion of a system processing information across time equal to the history length of the CNN. We were therefore wondering whether we could extract receptive fields from the gradient $J$ and Hessian $H$ of the CNN in a manner similar to how they can be derived from the first- and second-order Volterra kernels ($k_1$ and $k_2$ ; see 'Methods') (*Marmarelis, 1997*; *Marmarelis, 2004*; *Mitsis et al., 2007*). To this end, we simulated a system that uses two parallel receptive fields to process an input. The filtered responses are subsequently passed through two differently structured nonlinearities to yield the output (*Figure 3A* and 'Methods'). Notably, due to the structure of the nonlinearities, we expect the first receptive field (here called 'linear receptive field') to be equivalent to $J/k_1$ and the second receptive field (here called 'nonlinear receptive field') to appear as an eigenvector of $H$ and $k_2$.

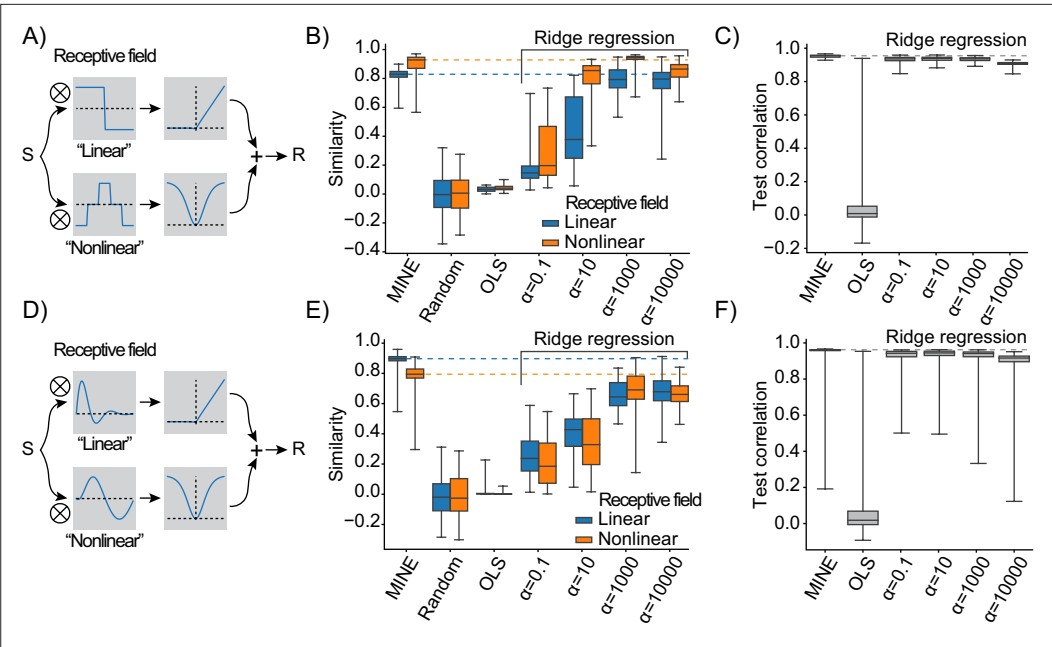

**Figure 3.** Model identification of neural encoding (MINE) characterizes linear and nonlinear receptive fields. (**A**) Schematic of the test response generation. Inputs S (either white-noise or slow fluctuating) are convolved in parallel with two receptive fields acting as filters. The result of one convolution is transformed by an asymmetric nonlinearity (top), the other through a symmetric one (bottom). The results of these transformations are summed to create the response R that is a stand-in for a neural response that depends on one linear and one nonlinear receptive field. (**B**) When presenting a slowly varying stimulus, the quality of receptive fields extracted by MINE (expressed as the cosine similarity between the true receptive fields and the respective receptive fields obtained by the analysis), as well as direct fitting of first- and second-order Volterra kernels through linear regression (OLS) as well as Ridge regression. Listed $\alpha$ indicates the strength of the Ridge penalty term. Linear receptive field blue, nonlinear orange. Dashed lines indicate median cosine similarity of the receptive fields extracted using MINE. (**C**) Correlation to validation data presented to the fit models as a means to assess generalization. Dashed lines indicate correlation to test data of the MINE model. (**D–F**) Same as (**A–C**) but with smoothly varying receptive fields. All data is across 100 indepedent simulations.

The online version of this article includes the following figure supplement(s) for figure 3:

**Figure supplement 1.** Model identification of neural encoding (MINE) characterizes linear and nonlinear receptive fields.

As a comparison, we used regression to directly fit the Volterra kernels (see 'Methods'). Notably, just like MINE, apart from the truncation of the series after the second term, the Volterra analysis is highly flexible since most scalar-valued functions can be approximated using an infinite Volterra series (**Volterra, 1959**). System identification often employs specifically designed stimuli such as Gaussian white noise (**Marmarelis and Marmarelis, 1978**; **Korenberg and Hunter, 1990**; **Rieke et al., 1999**; **Schwartz et al., 2006**; **Gollisch and Meister, 2008**), which are difficult to realize in practice and severely restrict experimental conditions. Nonetheless, we first benchmarked both approaches using Gaussian white noise as input (**Figure 3—figure supplement 1A**). As expected, both MINE and directly fitting the Volterra kernels yielded receptive fields that were nearly indistinguishable from the ground truth (**Figure 3—figure supplement 1B**). This demonstrates that our simulation indeed results in receptive fields that can be discovered using MINE or a second-order Volterra model. However, for the analysis to be useful it is critical that receptive fields can be extracted on arbitrary and slowly varying stimuli as expected, for example, during naturalistic behavior. We therefore repeated the analysis using slowly varying stimuli (**Figure 3—figure supplement 1A**). Under these more naturalistic conditions, MINE still yielded well-fitting receptive fields (**Figure 3B** and **Figure 3—figure supplement 1C and D**). An ordinary regression fit of the Volterra kernels, on the other hand, failed to recover the receptive fields (**Figure 3B**), which is in line with the observation that ANNs form an efficient route to Volterra analysis (**Wray and Green, 1994**). We assumed that this failure was due to a lack of constraints, which meant

that the ordinary regression model could not handle the departure from Gaussian white noise stimuli. We therefore refit the Volterra kernels using Ridge regression (*Hastie et al., 2009*) with increasing penalties. Indeed, for high penalties, the direct fit of the Volterra kernels yielded receptive fields of almost comparable quality to MINE (*Figure 3B* and *Figure 3—figure supplement 1E–H*). The CNN model fit by MINE also had slightly higher predictive power compared to the best-fit Volterra model as indicated by higher correlations of predicted activity on a test dataset (*Figure 3C*). Repeating the same analysis with a second set of filters yielded similar results: MINE proved to be a slightly superior approach to extract the receptive fields (*Figure 3D–F*). This is not to say that no other way could be found to extract the receptive fields, but it underlines the fact that MINE is a practical solution to this problem, which yields high-quality receptive fields in addition to other information about the relationship between predictors and neural responses.

We used the same simulation to assess how the predictive power of the different models (MINE and Ridge regression), as well as the quality of extracted receptive fields, depends on the amount of training data. As expected, predictive power on a validation set increases with the number of training samples for all methods. The Ridge model with the lower penalty outperforms MINE for lower numbers of training samples (*Figure 3—figure supplement 1I*). However, this model does not yield usable receptive fields and in fact both Ridge models show a deterioration of extracted receptive fields for large numbers of training samples (*Figure 3—figure supplement 1J*). This is likely a result of the competition between the regularization penalty and the overall error of the fit. MINE appears more robust to this effect, but for very long sample lengths it may be required to adjust the CNNs regularization as well (*Figure 3—figure supplement 1J*).

Since Ridge regression constrains linear regression models, we were wondering how the effective degrees of freedom of the CNN and the different models would compare. Interestingly, in spite of having nearly 14,000 parameters, the effective degrees of freedom of the CNN model are <50 and the Ridge regression models that are successful in identifying the receptive fields approach similar effective degrees of freedom (*Figure 3—figure supplement 1K*). This is in line with successful approaches to system identification employing constraints such as Laguerre basis functions (*Friston et al., 1998*; *Marmarelis, 2004*; *Mitsis et al., 2007*). In the case of the CNN used by MINE, the effective degrees of freedom are limited by the L1 penalty on the weights (sparsity constraint), the Dropout, and the limited number of training epochs (*Figure 3—figure supplement 1L*).

Overall these results suggest that MINE can recover the types of receptive fields of neurons that can be obtained with system identification approaches under a broader set of biologically relevant stimulus conditions.

## Taylor analysis identifies predictors driving the response

Compared to regression models, the CNNs seemingly have a drawback: a lack of interpretability. Statistical methods can identify the factors that significantly contribute to a regression model's output. Similarly, the magnitude of individual weights in models fit to data can give an idea of the importance of specific predictors. Corresponding overt parameters do not exist in the CNN model. In principle, the extracted receptive fields could be used to identify contributing predictors since it would be expected that they are 'unstructured' for unimportant inputs. However, what to consider as 'unstructured' is not clearly defined. Theoretically, it would be possible to refit successful CNN models in a stepwise manner, leaving out or adding in specific predictors (*Benjamin et al., 2018*). However, since we are interested in uncovering the relationships between predictors, this could only succeed if all individual combinations between inputs are tested. This would be prohibitive for large sets of sensory and behavioral predictors as it would require repeatedly retraining all networks.

We therefore again utilized Taylor expansion. To account for the relationships that cannot be sufficiently explained by the second-order Taylor expansion shown in *Figure 2A*, we perform local expansions at various points instead of one expansion around the data average (*Figure 4A*). This allows for variation in the network gradient $J$ and Hessian $H$ in cases of high computational complexity. Since the Taylor decomposition is a sum of terms that depend on individual inputs or their combinations, we can use it to determine how important individual predictors, such as sensory stimuli, are in shaping the neural response (*Figure 4A* and *Figure 4—figure supplement 1A*). Across our ground-truth dataset, we find a high correlation between the change in output of the Taylor expansion and the true change in network output (*Figure 4B*, left panel), indicating that the local expansions using

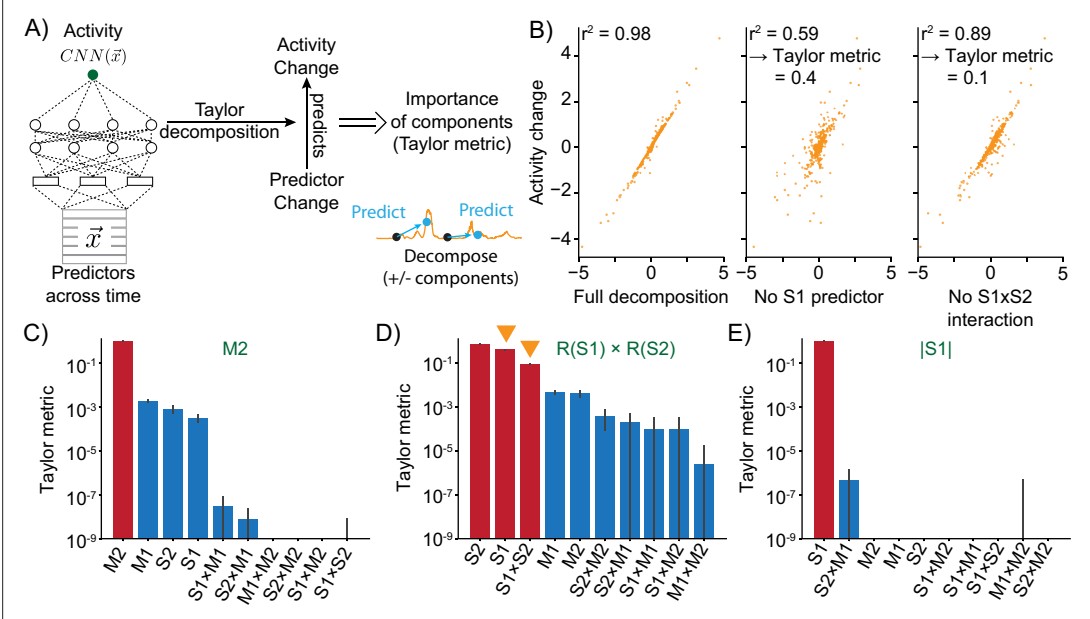

**Figure 4.** Taylor decomposition reveals contributing single factors and interactions. (**A**) The neural model translates predictors into neural activity. By Taylor decomposition, the function implemented by the convolutional neural network (CNN) can be linearized locally. Relating changes in predictors to changes in activity for full and partial linearizations reveals those predictors and interactions that contribute to neural activity. (**B**) Example of Taylor metric computation. Left: relationship between the CNN output and the full Taylor approximation. Middle: after removal of the term that contains the S1 predictor. Right: after removal of the term that describes the interaction between S1 and S2. (**C–E**) Three example responses and associated Taylor metrics. Red bars indicate predictors that are expected to contribute, blue bars those that should not contribute. Error bars are 95% bootstrap confidence intervals across N=20 independent simulations. (**C**) M2 response type. (**D**) R(S1) × R(S2) response type. Arrowheads indicate the metrics that are shown in the example (right and middle) of (**B**). (**E**) Response type encoding the absolute value of S1.

The online version of this article includes the following figure supplement(s) for figure 4:

**Figure supplement 1.** Taylor decomposition reveals contributing single factors and interactions.

first- and second-order derivatives sufficiently capture the relationship between predictors and neural activity. Importantly, decomposition is a way to efficiently test the importance of different predictors in contributing to network output and hence neural activity. We calculate a 'Taylor metric' score, which measures the fraction of explained variance of the response that is lost when terms are removed from the full Taylor expansion (*Figure 4B*).

On our ground-truth dataset, the Taylor metric correctly identifies the contributing predictors and their interactions. Sorting individual terms by this metric consistently ranks those that we expect to contribute (*Figure 4C–E*, red bars) higher than those that should not (*Figure 4C–E*, blue bars). This is true both for individual predictors and interaction terms in the case where the response depends on the product of inputs (*Figure 4C–E* and *Figure 4—figure supplement 1B-H*).

In summary, MINE was able to correctly identify contributions of predictors, such as sensory stimuli or behavioral actions, to neural responses by local expansions of the trained CNNs. MINE also correctly identifies nonlinear interactions in generating the neural responses on our ground-truth dataset. This indicates that we can further reduce the lack of interpretability of the CNN models and approach the expressivity of linear regression models while maintaining the ability to model nonlinear transformations and interactions of task variables.

## MINE characterizes cortical sensorimotor processing

Encouraged by MINE's ability to identify responses related to behavior and stimuli and its ability to characterize the nature of that relationship, we wanted to test the method on biological data. We applied MINE to a publicly available widefield calcium imaging dataset recorded in the mouse cortex (*Musall et al., 2019a*). The dataset consists of 22 task-related predictors (stimuli, reward states, instructed, and noninstructed movements) and calcium activity time series across 200 temporal components that were used to compress each session's widefield imaging data *Musall et al., 2019a* from 13 mice. It

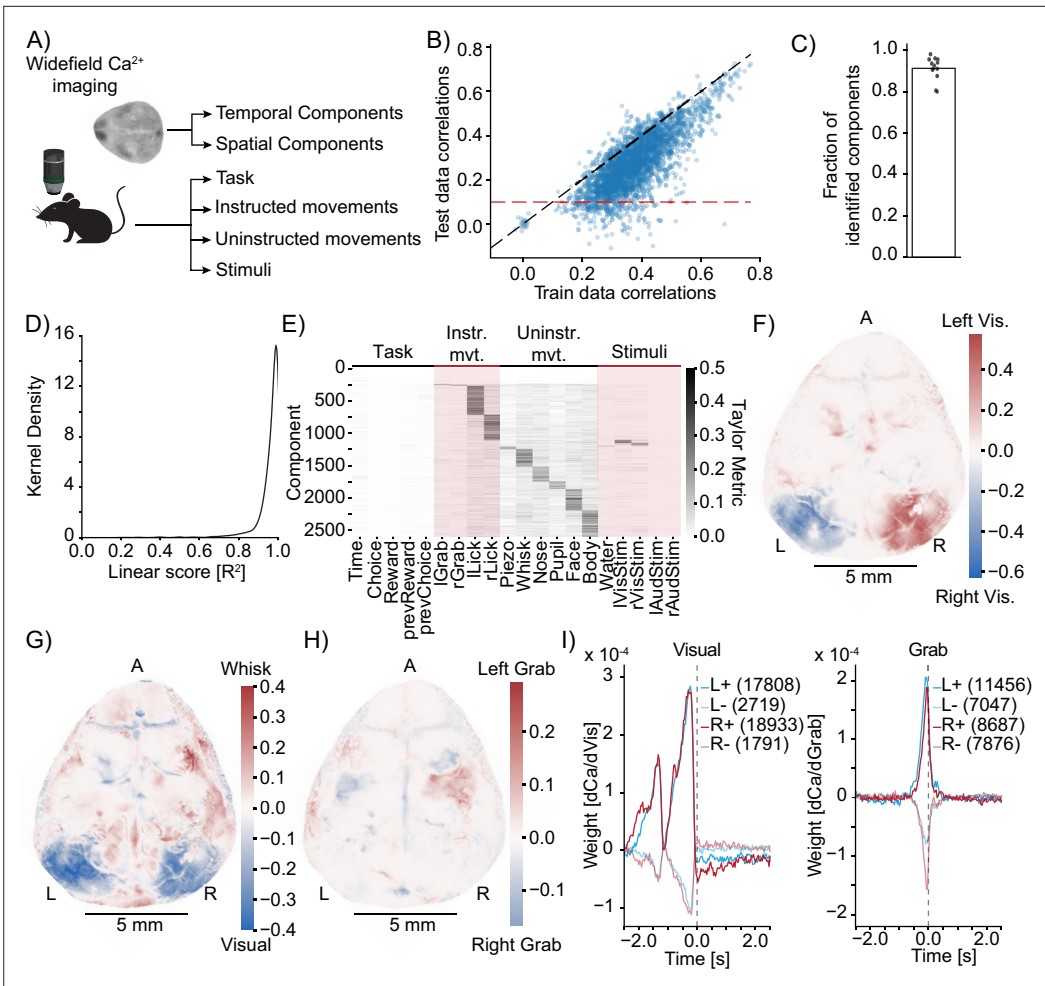

**Figure 5.** Model identification of neural encoding (MINE) identifies cortical features of sensorimotor processing during a learned task. (**A**) Simplified schematic of the widefield imaging experiment conducted in *Musall et al., 2019a*. (**B**) MINE test data vs. training data correlations on 200 temporal components from 13 sessions across 13 mice. Black dashed line is identity, red dashed line is the test correlation threshold to decide that a component had been identified by MINE. (**C**) In each of the 13 sessions, the fraction of identified components (dots) as well as the average (bar). (**D**) Across 200 components each in the 13 sessions, the distribution of the linear score computed by MINE (coefficient of variation for the truncation of the Taylor expansion after the linear term as in *Figure 2*). (**E**) Across all components from all 13 sessions that have been identified by MINE, the Taylor metrics that were significantly larger than 0. Components (rows) have been sorted according to the predictor with the maximal Taylor metric. (**F**) Per-pixel Taylor metric scores for the right visual stimulus ('rVisStim') subtracted from those of the left visual stimulus ('lVisStim'). A, anterior; L, left; R, right. (**G**) As in (**F**) but the sum of the visual stimulus Taylor metrics ('lVisStim+rVisStim') has been subtracted from the Taylor metric of the whisking predictor ('Whisk'). (**H**) As in (**F**) but the Taylor metric for the right grab predictor ("rGrab") has been subtracted from the Taylor metric of the left grab predictor ('lGrab'). (**I**) Clustering of per-pixel receptive fields, separating excitatory (bold lines) from inhibitory responses (pale lines). Pixels were selected to only include the top 10% Taylor metrics for visual (left plot) and grab (right plot) predictors; left blue, right red. Numbers in parentheses indicate cluster size. The gray dashed line indicates time 0, that is, the time at which calcium activity is measured. Note that the sensory receptive field (visual) is exclusively in the past (future stimuli do not drive the neurons) while the motor receptive fields (grab) slightly overlap with the future, indicating that neurons ramp up activity before the behavioral action.

The online version of this article includes the following figure supplement(s) for figure 5:

**Figure supplement 1.** Model identification of neural encoding (MINE) identifies cortical features of sensorimotor processing during a learned task.

had previously been analyzed using linear regression (*Musall et al., 2019a*). We analyzed one session from each of the 13 mice present in the dataset with MINE (*Figure 5A*). As in the ground-truth data, we split the data into a training set (first two-thirds of the session time) and a test-set (last third). To be able to capture not only the activity caused by past stimuli but also the preparatory activity leading up to movements, we shifted the predictor traces with respect to the activity trace (see 'Methods'). As expected, correlations between MINE predictions and true activity on the test dataset are overall smaller than on the training set; however, in the majority of cases the CNN generalizes well to the new data (*Figure 5B*). Given that many individual neurons contribute to each pixel and temporal component within this dataset, and that a lot of brain activity is likely unrelated to the behavioral task, we expect that a large fraction of variance in each component is likely unexplainable by any model that does not take internal brain states into account. We therefore chose a lenient correlation cutoff of $r = 0.1$ for the test data (red line in *Figure 5B*) to decide that a component had been successfully fit by MINE (*Figure 5B and C*). On average, this led to the identification of >91% of all components per session (*Figure 5C*) on which Taylor metric and complexity were computed. Notably, MINE assigns low complexity to the majority of components; only 3% of fit components have a linear approximation score <0.8. This means that the relationships between predictors and neural activity are largely linear (*Figure 5D*). While this may seem surprising given the high nonlinearity of cortical processing, it is likely caused by the low resolution of the data. Each component blends the activity of hundreds of thousands of neurons. Averaging across many individual neurons that each may have their own nonlinear responses likely obfuscates nonlinear effects.

Across all sessions, we found a broad representation of predictors (*Figure 5E*). As expected from the findings of *Musall et al., 2019a*, uninstructed movements appear overrepresented sporting the largest Taylor metric in more than half of the fit components. We next mapped the results of MINE back into acquisition space and recalculated the Taylor metrics for each imaging pixel for one example session. The obtained results further validate MINE's utility on biological data. Specifically, visual stimuli are shown to contribute most strongly to responses in visual cortical areas with the expected left–right asymmetry (*Figure 5F*). At the same time, comparing visual sensory responses to whisk responses shows that cortical regions enhanced for whisking (*Figure 5G*) correspond very well to regions marked strongly with a whisk event kernel in Figure 2 of *Musall et al., 2019a*. Furthermore, instructed left and right grab events largely contribute to motor cortical regions, again with the expected left–right asymmetry (*Figure 5H*). Repeating this procedure on all sessions (*Figure 5—figure supplement 1*) reveals general agreement. But we note that not all sessions seem to have regions that are as clearly related to the chosen inputs according to Taylor analysis, which is especially apparent for left and right instructed grabs (*Figure 5—figure supplement 1C*).

We next sought to determine the receptive fields that govern stimulus processing in individual pixels (*Lehky et al., 1992*; *Dayan and Abbott, 2001*). We extracted the receptive fields across pixels that are strongly related to either left/right visual stimuli or left/right instructed grabs. We broadly clustered the receptive fields into two groups (see 'Methods') to separate excitatory and inhibitory effects. Receptive fields for left and right visual stimuli in contralateral brain regions are highly similar to each other (*Figure 5I*, left). The excitatory effect is stronger than the inhibitory effect (larger coefficients in the positive receptive fields) and both are clearly biphasic. This indicates that the events around the time of the stimulus as well as $\sim 1.5\,\mathrm{s}$ in the past strongly influence the activity of the visual neurons. We note that this biphasic structure mimics the linear regression kernel in Figure 3b of *Musall et al., 2019a*. The visual receptive fields have essentially zero weight after the current time, which is expected since future stimuli are unlikely to influence current neural activity. The receptive fields of left and right grab neurons, on the other hand, are much sharper, indicating that these neurons influence movement over short timescales (*Figure 5I*, right). Furthermore, the grab-related receptive fields contain coefficients different from baseline for up to 100 ms into the future. This suggests preparatory activity, that is, that future movements are reflected in current neural activity.

In summary, the results presented above demonstrate the applicability of our method to biological data and the potential for identifying diverse feature sets of sensorimotor processing.

## Functional characterization of thermoregulatory circuits

Encouraged by MINE's performance on ground-truth and mouse cortical data, we sought to use it to gain novel insight into zebrafish thermoregulatory circuits. Temporal transformations of stimuli are a

notable feature of how zebrafish process temperature information. In previous research, we identified neurons and centers in the larval zebrafish brain that process temperature: specifically, neurons that compute the rate of change of the temperature stimulus (*Haesemeyer et al., 2018*) as well as neurons whose activity is consistent with integration of temperature fluctuations (*Haesemeyer et al., 2019*). Due to the nature of these transformations, which were unknown a priori, simple regression-based approaches failed to identify neurons involved in temperature processing. We therefore previously resorted to clustering to identify these neurons. However, because behavior is stochastic, one particular drawback of clustering was that it precluded identifying neurons that integrate thermosensory stimuli and behavioral actions. Since MINE can learn and capture both temporal transformations and interactions, we set out to gain deeper insight into thermoregulatory processing. To this end, we used MINE to identify and characterize neurons during a thermosensory task, revealing predictors contributing to their activity, extracting their receptive fields and characterizing their computational complexity.

To classify neurons by clustering either requires presenting the same stimulus to every animal and neuron while imaging or it requires a strategy of clustering event-triggered activity. With MINE we do not need to enforce any stimulus constraint. As a result, we were able to probe a wider variety of stimulus conditions. We imaged a total of 750 planes across 25 larval zebrafish that expressed the nuclear calcium indicator H2B:GCaMP6s *Freeman et al., 2014* in all neurons and the excitatory neuron marker vglut2a-mCherry *Satou et al., 2013* in presumed glutamatergic neurons. This dataset provided slightly more than fourfold coverage of the larval zebrafish brain. On each imaging plane, we presented a heat stimulus generated by randomly composing sine waves with frequencies between 0.005 Hz and 0.075 Hz (*Figure 6A* and *Figure 6—figure supplement 1A*). We concurrently recorded elicited tail motion at 250 Hz. We segmented the imaging data using CalmAn (*Giovannucci et al., 2019*), which identified $\sim 433{,}000$ active neurons across our dataset. From the tail motion data, we extracted (1) swim starts and (2) tail features that correlate with swim displacement ('vigor') and turn angle ('direction') in free swimming experiments (*Figure 6A* and *Figure 6—figure supplement 1C*).

We used the information about stimulus and elicited behaviors across time as inputs to MINE fitting CNNs to each neural calcium response (*Figure 6B and C*). The initial two-thirds of time served as training data and the last third as a test/validation set. Notably, due to the random nature of both our stimulus and the elicited behavior, test and training data were largely uncorrelated (*Figure 6—figure supplement 1B*). Predictive power over the test data therefore indicates that the neural network model generalizes and truly captures how stimulus and behavioral data are related to neural activity. We chose 50% of explained variance on the test data, corresponding to a correlation of $r = \sqrt{0.5}$, as a stringent cutoff to determine whether a neuron can be modeled by MINE. A considerably higher threshold was used on this dataset compared with the cortical data since we recorded single-cell activity. It is therefore less likely that activity of a true responder is mixed with unrelated background activity. Using cyclic permutations of the data as a control revealed that this cutoff corresponds to a 93-fold enrichment over control data (*Figure 6—figure supplement 1D*).

As a comparison, we fit a linear regression model to this dataset. The inputs included time-shifted versions of stimulus and behavioral variables to allow the model to learn temporal transformations *Musall et al., 2019a* and thereby put it on the same footing as the CNN model (*Figure 6C*). We used Ridge regression *Hastie et al., 2009* to improve generalization of the linear model. At the designated cutoff, a total of $\sim 42{,}000$ neurons were identified using either MINE or regression. 40% of these neurons, however, were exclusively identified by MINE (*Figure 6D*), indicating the superiority of the model-discovery approach. In fact, MINE consistently identifies more neurons regardless of the cutoff imposed on test data (*Figure 6—figure supplement 1E*).

We used MINE to assign the identified neurons to functional types according to (1) whether their encoding of stimulus or behavioral features is linear or nonlinear (computational complexity) and (2) which stimulus and behavioral features drove their activity (Taylor analysis). As before, we determined nonlinearity when the truncation of the Taylor expansion after the linear term did not explain at least 80% of the variance of the CNN output. For the Taylor analysis, we chose a stringent cutoff, requiring the Taylor metric to be significantly >0.1 with a p-value < 0.05 on the whole dataset (Bonferroni correction; effective p-value $\sim 1.26 \times 10^{-6}$), to determine whether a predictor influenced the activity of a neuron. Since interaction terms had very small Taylor metrics, we ignored them for neuron classification purposes. In total, we identified 33 functional neuron classes all of which were identified

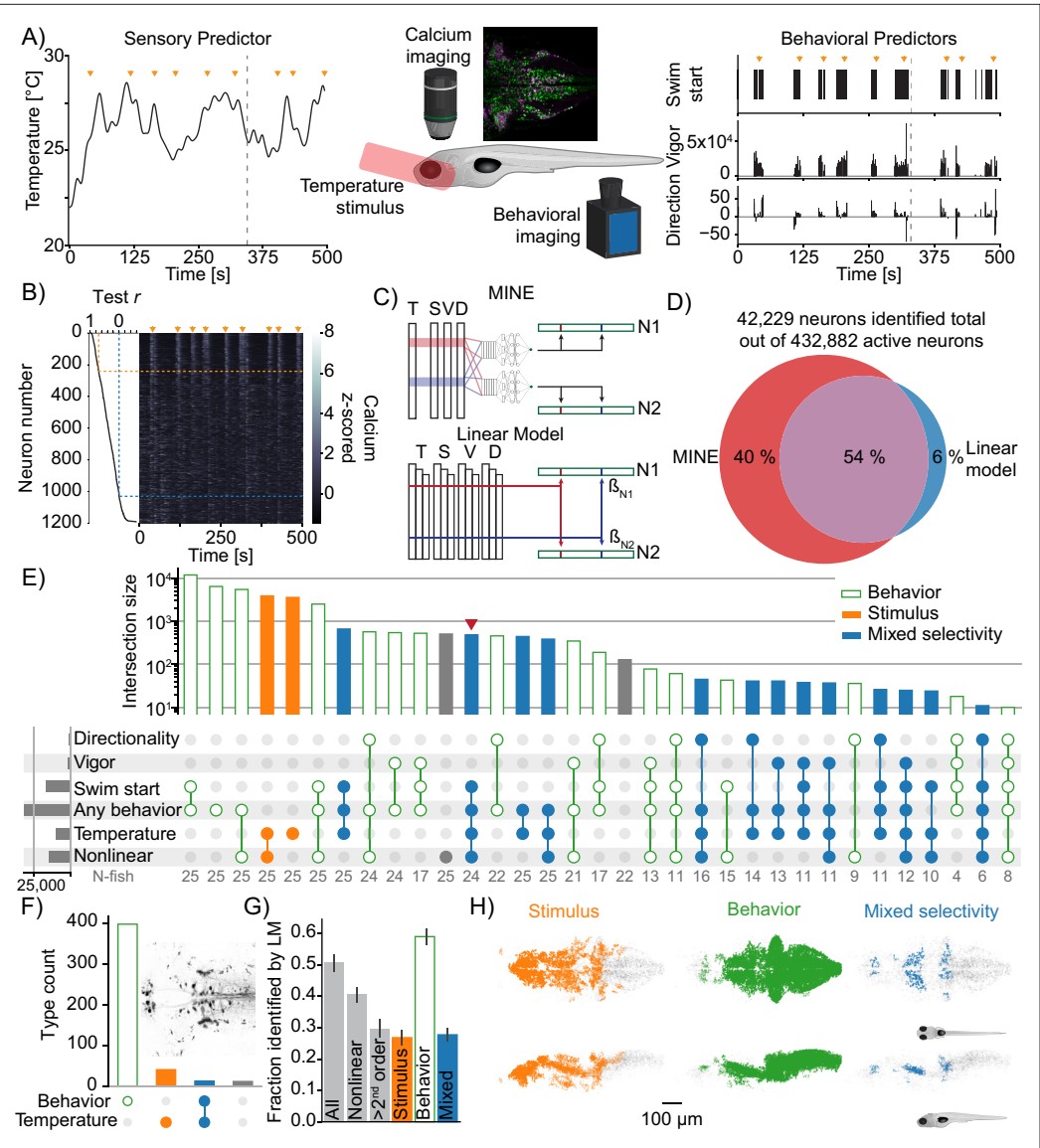

**Figure 6.** Using model identification of neural encoding (MINE) to probe larval zebrafish thermoregulatory circuits. (**A**) Experimental design. Larval zebrafish expressing nuclear GCaMP6s in all neurons and mCherry in glutamatergic neurons are imaged under a two-photon microscope while random heat stimuli are provided with a laser and behavior is inferred through tail motion, middle (inset shows a sum projection through the example trial depipcted on left and right and in (**B**) - edge-length = 400 microns). Left: example temperature trajectory during one trial. Right: behavioral responses recorded during the same trial. (**B**) Deconvolved calcium traces of all neurons identified in the plane imaged during the same trial as in (**A**) (heatmap), sorted by the test correlation achieved by the convolutional neural network (CNN) (plot on the left). Orange arrowheads mark the same timepoints as in (**A**) and (**B**). Orange dashed line indicates the fit cutoff used for deciding that a neuron was identified by MINE, blue line marks Pearson correlation of 0. (**C**) Illustration comparing MINE to the linear regression model. (**D**) Venn diagram illustrating fractions of CaImAn extracted neurons identified by MINE, the comparison LM model or both. (**E**) Plot of functional classes identified by Taylor analysis across 25 fish. Barplot at the top indicates total number of neurons in each class on a logarithmic scale. Dotplot marks the significant Taylor components identified in each functional class. Classes are sorted by size in descending order. Horizontal barplot on the right indicates the total number of neurons with activity depending on a given predictor. Orange filled bars mark classes only driven by the stimulus, green open bars those only driven by behavioral predictors while blue bars mark classes of mixed sensorimotor selectivity. Gray numbers in the row labeled 'N-fish' indicate the number of fish in which a given type was identified. The red arrowhead marks the functional type that is analyzed further in *Figure 7D*. (**F**) Classification of neurons labeled by reticulospinal backfills (inset shows example labeling) across six fish. Orange filled bars

*Figure 6 continued on next page*

*Figure 6 continued*

mark classes only driven by the stimulus, green open bars those only driven by behavioral predictors while blue bars mark classes of mixed sensorimotor selectivity. (**G**) For different functional neuron classes identified by MINE, the fraction also identified by the linear comparison model. Error-bars are bootstrap standard errors across N=25 zebrafish larvae. (**H**) Anatomical clustering of stimulus driven (left), behavior driven (middle), and mixed-selectivity (right) neurons. Neurons were clustered based on spatial proximity, and clusters with fewer than 10 neurons were not plotted (see 'Methods'). Asymmetric patterns for lower abundance classes likely do not point to asymmetry in brain function but rather reveal noise in the anatomical clustering approach.

The online version of this article includes the following figure supplement(s) for figure 6:

**Figure supplement 1.** Using model identification of neural encoding (MINE) to probe larval zebrafish thermoregulatory circuits.

across multiple fish (*Figure 6E*). As a control, we labeled reticulospinal neurons in six fish via spinal backfills and as expected the vast majority of these neurons were classified by MINE to be driven by behavioral features (*Figure 6F*). Across the brain, we identified multiple functional classes of neurons with mixed selectivity, that is, neurons with activity jointly driven by the temperature stimulus and behavioral outputs (*Figure 6E*, blue filled bars). These mark a previously unidentified class of neurons in the thermoregulatory circuit that might play a key role in behavioral thermoregulation: they could allow the animal to relate behavioral output to temperature changes and thereby characterize the thermal landscape. Analyzing the success of the linear comparison model according to functional class revealed a bias toward the identification of behavior-related activity and as expected low computational complexity. Stimulus or mixed-selectivity neurons, on the other hand, were underrepresented in the pool identified by the linear model (*Figure 6G*). This points to potential problems of bias when identifying neurons by means of such linear models.

We registered all our imaging data to a standard zebrafish reference brain (Z-Brain; *Randlett et al., 2015*). This allowed us to assess the distribution of identified functional neuron types throughout the brain using two complementary methods. We performed anatomical clustering (see 'Methods') to visualize regions with high densities of each functional type (*Figure 6H*) and determined in which annotated anatomical regions a functional neuron type is enriched (*Figure 6—figure supplement 1G–I*). While Stimulus-, Behavior- and Mixed-selectivity neurons are generally broadly distributed, they are enriched in specific brain regions (*Figure 6H* and *Figure 6—figure supplement 1G-I*). We found stimulus-driven neurons to be enriched in telencephalic regions, as well as the habenula and its output region the interpeduncular nucleus (*Figure 6—figure supplement 1G*). As expected, behavior-driven neurons are enriched in hindbrain regions and the midbrain nucleus of the medial longitudinal fasciculus (*Figure 6—figure supplement 1H*; *Severi et al., 2014*; *Thiele et al., 2014*). Mixed-selectivity neurons occupy some regions shared with stimulus-driven neurons such as telencephalic area M4 and the interpeduncular nucleus but are also strongly enriched in the raphe superior as well as the caudal hypothalamus and the torus longitudinalis (*Figure 6—figure supplement 1I*). Our current dataset did not cover sensory ganglia well (especially not the trigeminal ganglion) with the exception of the olfactory epithelium where we found temperature-sensitive neurons in a medial zone (*Figure 6—figure supplement 1I*). This is in line with reports in mice and frogs that describe specific thermosensitive neurons in the olfactory epithelium of these species (*Schmid et al., 2010*; *Kludt et al., 2015*; *Fleischer, 2021*). Overall these results demonstrate that functional cell types identified by MINE segregate spatially within the brain, likely forming organizational units within the thermoregulatory circuit.

## Computational features of thermoregulatory circuits

We next sought to use MINE to analyze computational features of thermoregulatory circuits. We subdivided neural classes according to the behavioral features they control, the sensory features they extract (their receptive fields), and their computational complexity. In the case of mixed-selectivity neurons, we used the predictive power of MINE to gain insight into how they integrate thermosensory and behavioral information. Mapping computational features to the zebrafish brain subsequently revealed the anatomical organization of computational features. Analyzing brain regions for enrichment of encoding different behavioral features suggests a segregation in the control of swim starts, swim speed (vigor), and turn angle (directionality). Even though behavior-related neurons are enriched

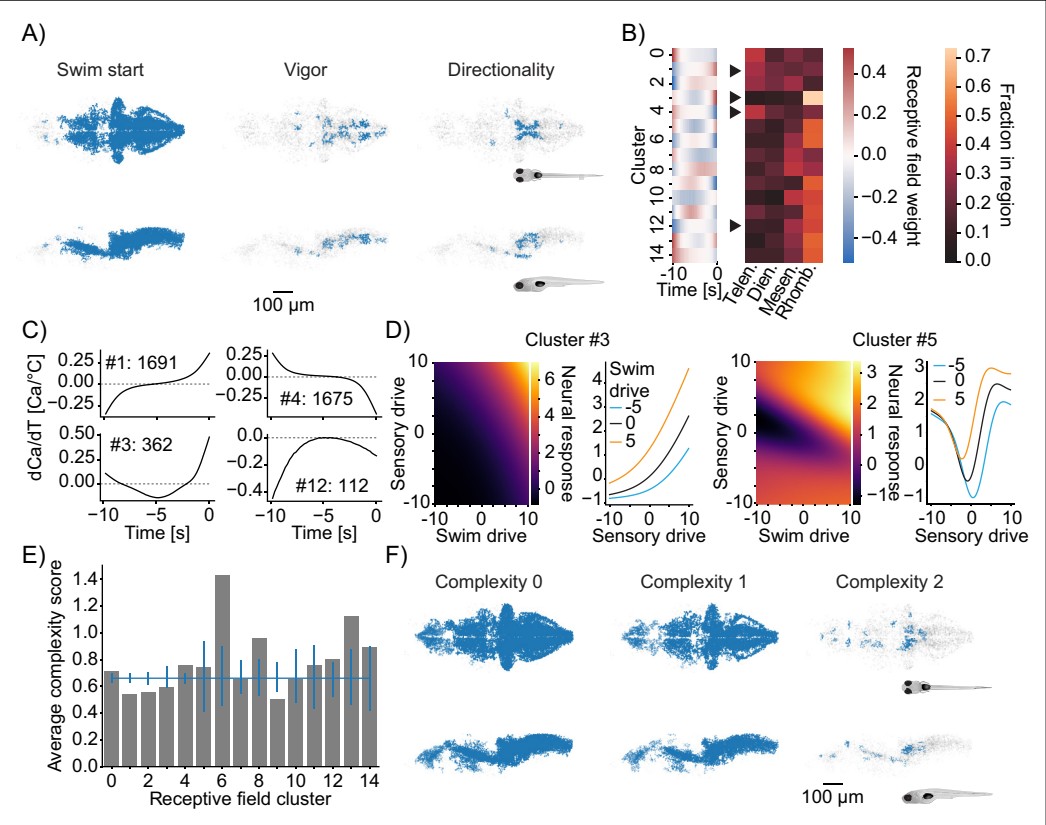

**Figure 7.** Functional subdivisions of thermoregulatory circuits. (**A**) Anatomical clustering of neurons encoding swim starts (left), swim vigor (middle), and swim direction (right). Neurons were clustered based on spatial proximity, and clusters with fewer than 10 neurons were not plotted (see 'Methods'). Asymmetric patterns for lower abundance classes likely do not point to asymmetry in brain function but rather reveal noise in the anatomical clustering approach. (**B**) Subclustering of stimulus-selective neurons according to their temporal receptive field (left heatmap). Right: heatmap visualizes for each cluster what fraction of that cluster is present in the major four subdivisions of the zebrafish brain (Telen., telencephalon; Dien., diencephalon; Mesen., mesencephalon; Rhomb., rhombencephalon). Arrowheads indicate differentially distributed example clusters highlighted in (**C**). (**C**) Temporal receptive fields of example clusters. Each plot shows the influence of a change in temperature at the indicated timepoint on the activity (as measured by calcium) of a neuron within the cluster. Numbers reflect the number of neurons present in each given cluster. Dashed lines indicate 0 where the number of 0-crossings of each receptive field indicate if the neuron responds to absolute temperature value (no crossings, cluster 12), to the first derivative (velocity of temperature, increasing, cluster 1; decreasing cluster 4) or to the second derivative (acceleration of temperature, cluster 3). (**D**) Exemplars of two clusters (full set in *Figure 7—figure supplement 1E*) of nonlinear mixed-selectivity neurons that integrate thermosensory information with information about swim start. Heatmaps show predicted neural calcium response for varying levels of swim- and sensory drive (see 'Methods'). Line plots show predicted calcium responses (Y-axis) to different sensory drives (X-axis) at different levels of swim drive (blue bar -5, black bar 0, orange bar +5). (**E**) Average complexity of each receptive field cluster shown in (**B**) (gray bars). Blue horizontal line reveals the total average complexity, and vertical blue lines indicate bootstrapped 95% confidence intervals around the average complexity based on the number of neurons contained within the cluster. If the gray bar is above or below that interval, the complexity within that cluster deviates significantly from the data average complexity. (**F**) As in (**A**) but clustering of neurons of complexity 0 (left), complexity 1 (middle), and complexity 2 (right).

The online version of this article includes the following figure supplement(s) for figure 7:

**Figure supplement 1.** Functional subdivisions of thermoregulatory circuits.

in the hindbrain overall (*Figure 6H*), subdividing the neurons reveals a broader distribution (*Figure 7A* and *Figure 7—figure supplement 1A–C*). Swim start neurons are enriched in the dorsal thalamus as well as the medial hindbrain, while the ventral thalamus, the forebrain, and the locus coeruleus preferentially encode swim speed (*Figure 7* and *Figure 7—figure supplement 1A and B*). Neurons that

encode turning are enriched in mid- and hindbrain regions (*Figure 7* and *Figure 7—figure supplement 1C*). Notably, these neurons could be involved in controlling these behavioral features or alternatively maintain efference copies of the motor commands for other purposes (*Odstrcil et al., 2022*).

To investigate the stimulus features that drive temperature encoding neurons, we performed clustering on their receptive fields that we retrieved using MINE as described above (*Figure 7B and C* and *Figure 7—figure supplement 1D*). A clustering approach based on the cosine similarity of receptive fields (see 'Methods') resulted in 15 clusters. For each of these clusters, neurons came from at least 15 fish across our dataset. Each cluster represents a functional neuron class that extracts specific features of the temperature stimulus (*Figure 7B*, left). Localizing different clusters in the brain reveals an anatomical organization of processing. Clusters 1 and 4 are enriched in the forebrain (*Figure 7B*, right), and their receptive fields suggest that they are most responsive to heating and cooling stimuli over slow timescales, respectively (*Figure 7C*). Their response to temperature increases or decreases is evidenced by the zero-crossing of their receptive fields while the slow rise/fall of the coefficients suggests computation over longer timescales. Cluster 3, on the other hand, computes the rate of heating over shorter timescales (*Figure 7C*) and is almost exclusively localized to the hindbrain (*Figure 7B*, right). Cluster 12 represents neurons that are excited by colder temperatures since the receptive field exclusively has negative coefficients (*Figure 7C*). This neuron type is found predominantly in hind- and midbrain regions (*Figure 7B*, right). We note that some of the uncovered receptive fields (e.g. clusters 1 and 4) have the largest departures from 0 at the start and end of the receptive fields. This might indicate that the chosen history length (here 10 s) is too short and does not cover the entire timescale of processing, which could be a result of the rather slow nuclear calcium indicator we chose for this study.

Mixed-selectivity neurons could provide zebrafish with important information about the thermal environment such as the slope of temperature gradients the fish can use to thermoregulate. We therefore wondered how these neurons combine thermosensory and behavioral information by visualizing response landscapes (akin to *Heras et al., 2019*). Here, we specifically focused on neurons that are driven by both swim starts and temperature inputs and that were classified as nonlinear (indicated by a red arrowhead in *Figure 6E*). We extracted the neurons' receptive fields for these quantities and analyzed predicted responses of the CNN fit by MINE to combinations of swim- and sensory drive. We defined 'drive' as scaled versions of the receptive field since it represents the ideal stimulus driving a neuron (*Dayan and Abbott, 2001*) (see 'Methods'). Clustering neurons according to these response landscapes (*Figure 7D* and *Figure 7—figure supplement 1E*) revealed diverse modes of integration across 10 clusters. Some neurons appear to add behavioral inputs linearly with thermosensory inputs (*Figure 7* and *Figure 7—figure supplement 1E*, clusters 8 and 9). Other neurons, however, show gating (*Figure 7D*, left, and *Figure 7—figure supplement 1E*, clusters 0 and 1). Here swim drive effectively scales the sensory responsiveness. Importantly, since these are temporal receptive fields, the swim drive input will scale with relative timing of swims, being the strongest when behavior occurs within the time window of the receptive field. These neurons could therefore act as coincidence detectors that inform larval zebrafish that temperature changes are linked to behavior as expected in a temperature gradient. We also found a cluster of neurons that display more complex integration of sensory and motor information (*Figure 7D*, right). At lower-than-average swim drive, these neurons symmetrically respond to both low and high sensory drive. However, at high swim drive, the response to positive sensory drive is enhanced, making the response of the neuron to temperature asymmetric in this regime.

Lastly, we sought to gain insight into computational features themselves, that is, those that occur between the sensory input and the observed activity of the neurons. Specifically, we used MINE to analyze the complexity of these computations according to truncations of the Taylor expansion after the linear and second-order terms (*Figure 2A*). We scored linear neurons as having complexity 0, those for which the linear truncation explains <80% of variance (nonlinear) but where the second-order expansion explains at least 50% of variance as having complexity 1 and the remainder requiring higher-order terms of the expansion as complexity 2. We found that complexity is mapped onto both functional (*Figure 7E*) and anatomical features (*Figure 7F*), indicating that this division is not arbitrary but rather indicates a meaningful difference in neuron function. Specifically, averaging the complexity score for different receptive field clusters reveals that three of these clusters have lower complexity than expected, indicating that most neurons in these clusters have responses that linearly depend

on the temperature stimulus (*Figure 7E*). One cluster (cluster 6) had an average complexity score >1, indicating that the majority of neurons in this cluster compute highly nonlinear transformations of the stimulus. Anatomical clustering of neurons by computational complexity reveals that different complexity classes are generally intermingled (*Figure 7G*). However, there appears to be some regionalization. Within the hindbrain, neurons of complexity class 2 are found medially in rhombomeres 1 and 2. We previously found that neurons in this region carry little information about ongoing behavior, which might indicate that they process thermosensory stimuli for other functions (*Haesemeyer et al., 2018*). Prominent clusters of complexity 2 neurons can also be found in the habenula and dorsal telencephalon consistent with the idea that these structures are involved in higher-order computations rather than the short-time control of thermoregulatory behaviors. Linear neurons, on the other hand, are enriched in the posterior and medial hindbrain.

In summary, these data reveal the power of MINE to identify computational features leading to a detailed classification of neurons into functional types. This automated, unbiased, and highly flexible analysis framework has the potential to greatly aid the analysis of large-scale neural data. To facilitate the adoption of MINE, we expose its functionality through a simple Python interface. This interface allows fitting the corresponding CNN to neural data and extracting all metrics (*Table 1*).

## Discussion

A common goal in analyzing neural data is to relate the activity of neurons to sensory stimuli, behavioral actions, internal states, or other features observed during ongoing behavior. To explore this connection, we neuroscientists often define a model dictating the structure of this relationship. When well thought-out, these defined models have the great advantage that their parameters can be interpreted in a biologically meaningful manner. However, even flexible models, if defined a priori, run the risk of not being able to match transformations occurring in the less than intuitive biological brain. Model-selection based on probabilistic programs overcomes some of these challenges, allowing for a model-free and flexible approach while maintaining predictive power (*Saad et al., 2019*). One drawback of this approach however is reduced interpretability of the nature of the relationship between inputs (in our case, stimuli, behaviors, etc.) and the output (in our case, neural activity). Here, we have presented MINE to overcome some of these challenges and offer an accessible framework for comprehensive analysis. Despite MINE's increased flexibility over predefined models, it maintains interpretability: through Taylor decomposition approaches, MINE (1) characterizes the computational complexity of neural processing giving rise to the activity of an individual neuron, (2) extracts linear and nonlinear receptive fields that define the ideal inputs driving a neuron, and (c) yields information about which predictors are most important in driving the neural response.

### Discovering encoding of information by neurons

Different paradigms are used to interpret neural activity. MINE specifically supports an encoding view, inspecting which information is encoded by the neurons of interest. ANNs have, however, also been used from a decoding perspective, probing what information can be gleaned about ongoing behavior from neural activity (*Frey et al., 2021*; *Schneider et al., 2022*). Here, we restricted the question of information encoding to individual neurons. Nonetheless, MINE could easily be extended to model joint population encoding. The most straightforward approach to this problem would be to decompose the population activity through methods such as principal or independent component analysis. These components would then be fit by MINE instead of single-neuron activity. Similarly, we only demonstrate the relationship of neural activity to external features (stimuli, behavioral actions) but neural population activity or local field potentials could be used as predictors. We currently do not exploit or characterize joint processing strategies. The efficiency of model-fitting could likely be improved by having the CNN model the relationships of multiple neurons (network outputs) to the same predictor inputs. Given the general applicability of the Taylor decomposition strategies, such networks could be analyzed in the same way as the current simpler CNN underlying MINE.

ANNs have been used extensively as models of brains to identify the principles underlying neural processing (*McClelland et al., 1987*). In the present work, CNNs exclusively serve as tools to find the relationships between features of interest and the activity of individual neurons. The network is a stand-in for the circuits transforming these features upstream (for sensation) or downstream (for

**Table 1.** The programmatic interface to model identification of neural encoding (MINE).
Details and example usage of the programmatic interface to MINE.

| class MineData | class Mine |
|---|---|
| *correlations_trained (n_neurons x 1)* | *train_fraction (float)* |
| Correlation of CNN prediction and activity on training portion of data | Which fraction of the data is used for training with remainder used for testing |
| *correlations_test (n_neurons x 1)* | *model_history (integer)* |
| Correlation of CNN prediction and activity on test portion of data | Number of timepoints the model receives as input |
| *taylor_scores (n_neurons x n_components x 2)* | *corr_cut (float)* |
| Taylor metric for each component (predictor and first-order interaction terms). The first entry along the last dimension is the mean score, the second entry is the bootstrap standard error. | If test correlation is less than this value for a neuron, it is considered 'not fit.' |
| *model_lin_approx_scores (n_neurons x 1)* | *compute_taylor (bool)* |
| Goodness of fit of linear Taylor model around the data mean to determine nonlinearity. | If true, compute Taylor metrics and complexity analysis (linear and second-order approximations). |
| *mean_exp_scores (n_neurons x 1)* | *return_jacobians (bool)* |
| Goodness of fit of second-order Taylor model around the data mean to derive complexity. | If true, return linear receptive fields. |
| *jacobians (n_fit_neurons x (n_timepoints x n_predictors))* | *taylor_look_ahead (integer)* |
| For each fit neuron, the receptive field of each predictor across time. | The number of timepoints to predict ahead when calculating Taylor metrics. |
| *hessians (n_fit_neurons x (n_timepoints x n_predictors) x (n_timepoints x n_predictors))* | *taylor_pred_every (integer)* |
| For each fit neuron, the matrix of second-order partial derivatives. Useful to extract second-order receptive fields. | Every how many frames a Taylor expansion should be performed to calculate Taylor metrics. |
| | Additional settable properties: |
| | *return_hessians (bool, default False)* If true, return matrices of second-order derivatives *model_weight_store (hdf5 file or group, default None)* If set, trained model weights for all models will be organized and stored in the file/group *n_epochs (integer, default 100)* The number of training epochs. |
| *save_to_hdf5(file_object, overwrite = False)* | *analyze_data(pred_data: List, response_data: Matrix) ->MineData* |
| Saves the result data to an hdf5 file or group | Takes a list of n_timepoints long predictors and a matrix of n_neurons x n_timepoints size and applies MINE iteratively to fit and characterize CNN relating all predictors to each individual neuron. |
| Example usage: | |
| predictors = [Stimulus, Behavior, State] | |
| responses = ca_data | |
| # NOTE: If predictors and response are not z-scored, | |
| # (mean = 0; standard deviation = 1) Mine will print | |
| # a warning | |
| miner = Mine (2/3, 50, 0.71, True, True, 25, 5) | |
| miner.model_weight_store = h5py.File("m_weights.h5", 'a') | |

*Table 1 continued on next page*

*Table 1 continued*

| class MineData | class Mine |
| --- | --- |
| result_data = miner.analyze_data (predictors, responses) | |
| all_fit = result_data.correlations_test >= 0.71 | |
| is_nonlinear = result_data.model_lin_approx_scores <0.8 | |
| is_stim_driven = (result_data.taylor_scores[:, 0, 0] – 3 x result_data.taylor_scores[:, 0, 1]) >0 | |

behavior), as well as the computational properties of the analyzed neuron. We are therefore interested in the properties of the whole CNN instead of its individual units. Extracting the CNN properties aims to characterize the neuron of interest in terms of how it encodes the features. Such an approach has been used previously to characterize neural receptive fields, recognizing the power of ANNs in capturing nonlinear transformations. For example, characterizations of nonlinear visual receptive fields (*Lehky et al., 1992*; *Lau et al., 2002*; *Prenger et al., 2004*; *Ukita et al., 2019*) and nonlinear auditory receptive fields (*Keshishian et al., 2020*). In retinal processing, works by Baccus and Ganguli *McIntosh et al., 2016*; *Tanaka et al., 2019* have used the increased flexibility of neural network models over classical linear–nonlinear models to characterize how retinal circuits process natural scene inputs. Internal neurons in the network matched retinal interneurons that were not used for data fitting and resulting models generalized better to novel data than linear–nonlinear models in spite of increased flexibility. Our comparison to Volterra analysis suggests that this might be because the effective degrees of freedom of CNN models are much lower than the total number of parameters but that the space in which these limited degrees of freedom act is better matched to neural data. We go beyond previous approaches in this study by combining modalities and providing an easily accessible framework for analysis of arbitrary neural data. MINE handles and disambiguates multiple different inputs originating from different domains (such as sensory stimuli and behavioral actions). MINE, furthermore, performs a more detailed characterization producing information about the linearity of the relationship of inputs and neural activity. These advances are made possible through mathematical analysis of the network.

## Characterizing the model implemented by the CNN

Different techniques aim at understanding how ANNs arrive at their outputs given a set of inputs (*Samek et al., 2019*). These techniques are generally referred to as 'explainable machine learning' and are particularly important in understanding what features classifiers use to make their decisions. These include visualization techniques as well as layer-wise relevance propagation (*Binder et al., 2016*) that aims to identify important input features driving the output of a classifier. In our case, however, we were interested to know which inputs persistently (across experimental time) are involved in driving the output of the network, that is, the activity of the modeled neuron. Notably, once the network is trained the different inputs are not separable anymore. This means that there is no guarantee that leaving out an unimportant input will not affect the output of the trained network. A workaround would be to iteratively train the network with subsets of inputs (*Benjamin et al., 2018*). However, this becomes infeasible even with a moderate number of inputs since all combinations would need to be tested. We therefore chose a continuously differentiable activation function for our network layers. This enabled Taylor decomposition to transform the network, point-by-point, into a separable sum of terms depending on individual inputs. This approach allowed us to infer which specific inputs drive the network output and hence the activity recorded in the modeled neuron. Notably this technique can also identify multiplicative interactions, which indicate coincidence detection. Yet, while such interactions were evident in our ground-truth dataset, we found little evidence of multiplicative interactions in the biological data. This is likely due to added noise obfuscating the contribution of these terms, which is comparatively small even on ground-truth data (*Figure 4D*). In fact, most information about a multiplicative interaction would be carried in the first derivative: namely the derivative with respect to one input would vary according to another input. While we did not directly exploit this insight in the present work, we do find evidence of multiplicative interactions in the mixed-selectivity neurons some

of which show gating of sensory responses by behavior (*Figure 7D* and *Figure 7—figure supplement 1E*).

To characterize the complexity of the relationship between features (stimuli, behavior, etc.) and neural activity, we analyzed truncations of the Taylor expansion of the network around the data average. While not guaranteed for all functions, we expect that the Taylor series converges for our network and hence that given enough terms the expansion would fully characterize the computation performed by the CNN. The ANN fit by MINE must encapsulate the transformations between the chosen features and the neural activity in order to be able to predict the activity. The classification of linear versus nonlinear processing by the network should therefore apply to the neural transformations as well. Especially in the absence of connectomics data, the metric of computational complexity can serve to group neurons into functional circuits. One might specifically expect that sensory processing proceeds from representations of lower complexity to those of higher complexity, that is, further departures from a simple encoding of stimuli. This type of extraction of more and more complex features is, for example, observed in cortical processing of visual stimuli (*Felleman and Van Essen, 1991*; *D'Souza et al., 2022*). We note, however, that computational complexity is not an absolute metric. Comparing the linearity for neurons responding to the same feature (e.g. a stimulus) is useful, but making the comparison across different features likely is not. In our zebrafish dataset, the majority of temperature-driven neurons were classified as nonlinear while most behavior-driven neurons were classified as linear. This, however, does not mean that the stimulus is processed less linearly than the behavior – it is likely a result of our stimulus feature being the raw temperature while our behavior features are removed from the nonlinear transformation that changes swim commands into appropriate tail motion.

We currently determine linear receptive fields (*Dayan and Abbott, 2001*; *Schwartz et al., 2006*) of fit neurons using the first-order derivatives of the network output with respect to the input. Similar to the analysis of higher-order Volterra kernels (*Marmarelis, 2004*; *Sandler and Marmarelis, 2015*), one could analyze eigenvectors of the matrix of second- or higher-order derivatives to get a deeper understanding of how this receptive field changes with stimuli in the case of neurons classified as nonlinear. Comparing direct fitting of Volterra kernels and extraction of first- and second-order kernels from MINE using Taylor analysis indeed highlights the feasibility of this approach (*Figure 3*). One crucial advantage of MINE is that it is considerably more robust to departures of stimuli from uncorrelated white noise with Gaussian distributed intensities, the ideal stimuli used for Volterra analysis (*Marmarelis and Marmarelis, 1978*; *Paninski, 2002*; *Marmarelis, 2004*; *Schwartz et al., 2006*). Such stimuli are impractical, especially when the goal is to study neural activity in freely behaving animals. However, our comparison to directly fitting the Volterra kernels also revealed that it is in fact possible to obtain the receptive fields this way. Volterra analysis, unlike MINE, could be fully unconstrained and therefore pose an advantage. While MINE is constrained by architecture and hyperparameters, our comparisons of extracting receptive fields with MINE versus an unconstrained fit of Volterra kernels revealed the limitations of not imposing any constraints (*Figure 3* and *Figure 3—figure supplement 1*). MINE offers a simple-to-use alternative that maintains flexibility while limiting effective degrees of freedom to allow for meaningful fits to real-world data. Independent of MINE, the structure of receptive fields is influenced by the acquisition modality. For our zebrafish data, we used a nuclear localized GCaMP6s. This variant has a long decay time (*Freeman et al., 2014*), and since MINE is fit on calcium data, effects of this decay time will appear in the extracted receptive fields. This likely explains why the receptive fields shown for the zebrafish data extend over many seconds. This issue could be overcome by better deconvolution of the imaging data which would require a higher acquisition framerate.

## Novel insight into biological circuits

Applying MINE to biological data revealed its potential to discover the relationships between stimuli and behaviors and neural activity across species and imaging modalities. On a mouse cortical widefield imaging dataset, MINE recovered previously published structure in the neural data. Task elements driving neural activity could be mapped to expected cortical regions and receptive field analysis revealed expected structure in visual receptive fields while the receptive fields associated with neurons encoding instructed movements showed preparatory activity. Performing two-photon calcium imaging in larval zebrafish while simultaneously presenting random temperature stimuli and recording behavior allowed a deeper dive into thermoregulatory circuits. MINE allowed us for the

first time to identify the neurons that integrate temperature stimuli and information about behavior. This was not possible previously since regression-based approaches failed to identify the neurons encoding temperature while clustering to identify these neurons relied on removing stochastic motor-related activity (*Haesemeyer et al., 2018*). Neurons integrating stimuli and behavior might serve an important role in thermoregulation. Specifically, they might allow larval zebrafish to estimate thermal gradients by comparing behavioral actions and their consequences in terms of changes in environmental temperature. While we have only begun to analyze the responses of these mixed-selectivity neurons, a simple clustering according to their response to combinations of sensory and motor drive revealed a variety of computed features (*Figure 7D* and *Figure 7—figure supplement 1E*). Combining receptive field analysis with the idea of computational complexity, on the other hand, provides a way to functionally subdivide temperature responsive neurons in great detail. This will greatly aid future studies trying to understand thermoregulatory circuits. Notably, the functional features we identified map to anatomical features in the brain pointing to structural subdivisions mirroring function. The ability to compare neurons, even though we used randomized stimuli, indicates another advantage of MINE. When recording neural activity during free exploration, across animals different neurons will be recorded under very different stimulus and behavior conditions. This means that their responses can usually be compared only by choosing different trigger events (certain behaviors, specific stimulus conditions) and analyzing neural activity around those. MINE, on the other hand, allows comparing networks fit on the whole activity time series removing the requirement and bias of choosing specific triggering events.

One surprising finding in comparing the cortical and zebrafish brain responses was the prevalence of nonlinear processing in zebrafish and the apparent lack of it in the cortical dataset. This is likely due to the different spatial resolution in the datasets. While the zebrafish data has single-neuron resolution, this is not the case for the cortical widefield data especially since we apply MINE not to individual pixels but activity components. Each of these components combines the activity of thousands of neurons. It is likely that this averaging obfuscates individual nonlinear effects.

In summary, MINE allows for flexible analysis of the relationship between measured quantities such as stimuli, behaviors, or internal states and neural activity. The flexibility comes from MINE turning analysis on its head: instead of predefining a model, MINE approximates the functional form that relates predictors to neural activity and subsequently discovers this functional form. This is possible since MINE does not make any assumptions about the probability distribution or functional form underlying the data it models. This allows MINE to identify more neurons than regression approaches on our zebrafish dataset (*Figure 6D*) while doing so in a less biased manner. The regression approach especially failed on stimulus-driven units, leading to an overrepresentation of behavior-related activity (*Figure 6G*). Testing how many neurons in our receptive field clusters were also identified by the linear model (*Figure 7* and *Figure 7—figure supplement 1D*) revealed that the linear model identified neurons belonging to each cluster. This argues that MINE increases sensitivity for identifying neurons, but at this level of analysis MINE did not necessarily identify functional types that were entirely missed by the linear model. However, in four clusters the linear model identified less than five neurons. This effectively means that the cluster would have been missed entirely when using the linear model data as the base for analysis. Accordingly, applying our clustering on the neurons identified by the linear model only recovers 4 instead of the 15 unique receptive field clusters that are obtained with MINE.

We propose that the fit CNN models the computation of upstream or downstream circuits of the neuron in question but we do not think or investigate whether it models the circuitry itself. This could be investigated in the future, but given the simplicity of the CNN used in MINE it is unlikely to provide interesting details in this manner. Instead, bringing more advanced network-wide analysis methods to bear on the fit CNN could reveal other interesting features of the computation 'surrounding' the neuron in question.

## Methods

Animal handling and experimental procedures were approved by the Ohio State University Institutional Animal Care and Use Committee (IACUC protocol # 2019A00000137 and 2019A00000137-R1).

## Design and training of convolutional neural network

A network that was simple, and therefore quick to train, but maintained expressivity was sought for this study and a three-layer CNN was chosen. We note that the input to the network is a chunk of time equal to the size of the convolutional layers. A similar architecture could therefore be realized without separate convolutional layers by simply expanding the number of input nodes. The first layer was built to contain 80 convolutional filters. We do not expect that the network will have to learn 80 separate filters; however, the advantage is that these filters contain a large amount of pre-training variation. This eases learning of temporal features important for the generation of the neural response. The convolutional layer was kept linear and followed by two 64-unit-sized dense layers with swish activation function, which subsequently fed into a linear output unit. This in effect means that each unit in the first dense layer performs a convolution on the input data with a filter that is a weighted average of the 80 convolutional filters.

The ground-truth dataset as well as part of a previous zebrafish dataset (*Haesemeyer et al., 2018*) were used to optimize the following hyperparameters: a sparsity constraint to aid generalization, the learning rate, the number of training epochs, and the activation function. The activation function was to be continuously differentiable, a requirement for subsequent analyses. Dropout was added during training to further help with generalization; however, the rate was not optimized and set at 50%. The history length (10 s) of the network input was set somewhat arbitrarily, but initial tests with shorter lengths led to problems of fitting the larval zebrafish dataset likely because of the slow calcium indicator time constant of nuclear GCaMP6s (*Freeman et al., 2014*). We note that we did not optimize the overall architecture of the network. This could certainly be done and all subsequent analysis methods are architecture agnostic as long as the network is a feed-forward architecture. *Related Python file in the repository: model.py.*

## Characterizing the linearity and complexity of the transformation

To compute measures of the complexity of the relationship between the predictors and neural activity, a Taylor expansion across the predictor average $\bar{x}$ was performed and the predictions of truncations of this Taylor expansion were compared to the true model output, sampling predictors every second. The Jacobian (vector of first-order derivatives of the output with respect to the input) and the Hessian (matrix of second-order derivatives of the output with respect to the input) were computed using Tensorflow. We note that since our network has a single output, the Jacobian is not a matrix but simply the gradient vector of the output with respect to the input.

The Jacobian and Hessian were used to compute the following first- and second-order approximations of the network output using the Taylor expansion:

$$\hat{f}_{\bar{x}}^{1st}(\vec{x}) = f(\bar{x}) + (\vec{x} - \bar{x})^T J(\bar{x}) \tag{1}$$

$$\hat{f}_{\bar{x}}^{2nd}(\vec{x}) = f(\bar{x}) + (\vec{x} - \bar{x})^T J(\bar{x}) + \frac{1}{2}(\vec{x} - \bar{x})^T \mathbf{H}(\vec{x} - \bar{x}) \tag{2}$$

These truncations were used to compute a 'linear approximation score' $LAS_f$ and a 'second-order approximation score' $SOS_f$ as the coefficients of determination quantifying the variance explained by the truncation of the true network output.

$$LAS_f = 1 - \frac{\sum_{\vec{x}}\left(f(\vec{x}) - \hat{f}_{\bar{x}}^{1st}(\vec{x})\right)^2}{var(f(\vec{x}))} \tag{3}$$

$$SOS_f = 1 - \frac{\sum_{\vec{x}}\left(f(\vec{x}) - \hat{f}_{\bar{x}}^{2nd}(\vec{x})\right)^2}{var(f(\vec{x}))} \tag{4}$$

We note that for a purely linear function we expect $LAS_f$ to be 1, and this score is therefore a measure of the linearity of the relationship between predictors and the neural response. $SOS_f$, on the other hand, quantifies how good a second-order model is in predicting the neural response, and we therefore use it to further define the complexity of the relationship between the predictors and the neural activity.

Based on ROC analysis on ground-truth data, $LAS_f$ was thresholded at $R^2 = 0.8$. Neurons for which the linear expansion explained <80% of variance of the true network output were considered nonlinear (see 'Results'). To assign complexity classes to zebrafish neurons, $SOS_f$ was thresholded at $R^2 = 0.5$; in other words, neurons for which the seccond-order expansion did not explain at least 50%

of the variance of the true network output were considered complex. Then neurons were assigned to one of three respective complexity classes: 0 for neurons for which the linear expansion $LAS_f$ indicated linearity, 1 if $LAS_f < 0.8$ and $SOS_f \geq 0.5$, or 2 if the neuron was deemed nonlinear and $SOS_f < 0.5$.

As a comparison, two metrics related to the nonlinearity of the function implemented by the CNN were calculated: a 'curvature metric' that quantifies the magnitude of the influence of the second-order derivative, and the 'nonlinearity coefficient' that approximates the error made by linearizing the network. It is important to note that given random inputs that are unrelated to the training data, it is very likely that the CNN will produce outputs that nonlinearly depend on these inputs. Both metrics are therefore calculated with data that forms part of the training manifold (*Raghu et al., 2023*). We also note that the 'curvature metric' does not in fact compute the curvature of the multidimensional function implemented by the CNN. Instead, the average magnitude of the vector induced by the Hessian on a unit magnitude step in the data space is used as a proxy. With $x$, $x'$, and $f$ defined as in 'Identifying contributing predictors by Taylor decomposition', the curvature metric is calculated according to

$$C_f = \frac{1}{n} \sum_{\vec{x}, \vec{x'}} \mathbf{H}(\vec{x})(\vec{x'} - \vec{x}) \tag{5}$$

The nonlinearity coefficient (NLC) was computed according to *Philipp and Carbonell, 2018* quantifying

$$NLC_{f,D} = \sqrt{\frac{\mathbb{E}_{\vec{x} \sim D}\left[Tr\left(J(\vec{x})^T Cov_{\vec{x}} J(\vec{x})\right)\right]}{Var_f}} = \sqrt{\frac{\mathbb{E}_{\vec{x}, \vec{x'} \sim D}\left[Tr\left(J(\vec{x})^T(\vec{x'} - \vec{x})(\vec{x'} - \vec{x})^T J(\vec{x})\right)\right]}{Var_f}} \tag{6}$$

where $D$ is the data distribution, $Tr$ is the trace operator, and $Cov_{\vec{x}}$ is the data covariance while $Var_f$ Varf is the output variance (since the output in the present case is a single value). The right form highlights that the NLC compares the variance of the linear approximation of the output ($J(\vec{x})^T(\vec{x'} - \vec{x})$) with the true output variance.

*Related Python files in the repository: taylorDecomp.py, perf_nlc_nonlin.py and utilities.py.*

## Comparison to Volterra analysis/determination of receptive fields

The discrete-time Volterra expansion with finite memory of a function up to order 2 is defined as

$$r(t) = k_0 + \sum_{\tau_1=0}^{T} k_1(\tau_1)s(t - \tau_1) + \sum_{\tau_1=0}^{T} \sum_{\tau_2=0}^{T} k_2(\tau_1, \tau_2)s(t - \tau_1)s(t - \tau_2) \tag{7}$$

where $r(t)$ is the response and $s(t)$ is the stimulus, and $k_0$, $k_1$, and $k_2$ are the zeroth-, first-, and second-order Volterra kernels, respectively.

This can be rewritten in matrix form where

$$\mathbf{k}_1 = \begin{vmatrix} k_1(0) \\ k_1(1) \\ \vdots \\ k_1(T) \end{vmatrix} \tag{8}$$

$$\mathbf{k}_2 = \begin{vmatrix} k_2(0,0) & k_2(0,1) \ldots k_2(0,T) \\ k_2(1,0) & k_2(1,1) \ldots k_2(1,T) \\ & \vdots \\ k_2(T,0) & k_2(T,1) \ldots k_2(T,T) \end{vmatrix} \tag{9}$$

$$\vec{s}(t) = \begin{vmatrix} s(t) \\ s(t-1) \\ \vdots \\ s(t-T) \end{vmatrix} \tag{10}$$

such that

$$r(t) = k_0 + \vec{s}(t)^T k_1 + \vec{s}(t)^T k_2 \vec{s}(t) \tag{11}$$

We note that since our network takes predictors across time as input, it can be easily seen that $\vec{x}$ is equivalent to $\vec{s}$ and therefore that the Taylor expansion (*Equation 2*) of the CNN is equivalent to the Volterra series with finite memory above (*Equation 11*). This indicates that there is a direct correspondence between the elements of the Taylor expansion and those of the Volterra expansion, where $J$ corresponds to $k_1$ and $H$ corresponds to $k_2$ and our CNN would be an effective means to fit the Volterra kernels (*Wray and Green, 1994*). This suggests that neural receptive fields can be extracted from $J$ and $H$ of the Taylor expansion in the same manner as they can be obtained from $k_1$ and $k2$ of the Volterra expansion.

To determine whether MINE could be used to extract system-level characteristics (such as receptive fields) akin to Volterra analysis, a model was used to turn a randomly generated stimulus into a response by passing the input through two orthogonal filters followed by two nonlinearities. The outputs of these two parallel stages were subsequently combined using addition to form the response. The stimulus was either generated as a Gaussian white noise stimulus (by drawing values $S_t \sim \mathcal{N}(0,1)$) or as a smoothly varying random stimulus as described in 'Ground-truth datasets.' The model consisted of two sets of orthogonal filters, $f_{linear}$ and $f_{nonlinear}$ of length 50 timepoints, as well as two nonlinearities $g_{linear}$ and $g_{nonlinear}$ (see *Figure 3*). The terms $f_{linear}$ and $f_{nonlinear}$ for the filters were chosen because the former is expected to be contained within $J/k_1$, that is, the linear term of the Volterra expansion while the latter is expected to be contained within $H/k_2$, that is, the quadratic term of the Volterra expansion. We note, however, that $f_{linear}$ can in addition be extracted from $H/k_2$ due to the particular function chosen for $g_{linear}$. Given a stimulus $s$, the response $r$ was constructed according to

$$r_{linear} = g_{linear}(s * f_{linear}) \tag{12}$$

$$r_{nonlinear} = g_{nonlinear}(s * f_{nonlinear}) \tag{13}$$

$$r = r_{linear} + r_{nonlinear} \tag{14}$$

, where * denotes convolution.

The two nonlinearities were constructed such that $g_{linear}$ was asymmetric and therefore led to a shift in the average spike rate, while $g_{nonlinear}$ was symmetric around 0 so that it predominantly influences the variance of the spike rate:

$$g_{linear}(x) = \frac{x}{1+e^{-5x}} \tag{15}$$

$$g_{nonlinear}(x) = (0.25 - \frac{e^x}{(1+e^x)^2}) * 15 \tag{16}$$

The constant multiplier in $g_{nonlinear}$ was chosen such that $r_{nonlinear}$ is in a similar range as $r_{linear}$.

The model therefore generates a response from which the two filters should be recoverable when fitting the first- and second-order Volterra kernels either through direct means akin to canonical system identification approaches or via the CNN used in MINE.

Given the Volterra kernels $k_0$, $k_1$, $k_2$ ($CNN(\vec{x})$, $J$, $H$), we attempted to recover $f_{linear}$ and $f_{nonlinear}$ by extracting the principal dynamic modes PDM from the kernels *Marmarelis, 1997* by performing eigen decomposition on an intermediate matrix $Q$, which combines all kernels in the following manner:

$$Q = \begin{vmatrix} k_0 & \frac{1}{2}k_1(0) & \frac{1}{2}k_1(1) \ldots \frac{1}{2}k_1(T) \\ \frac{1}{2}k_1(0) & k_2(0,0) & k_2(0,1) \ldots k_2(0,T) \\ \frac{1}{2}k_1(1) & k_2(1,0) & k_2(1,1) \ldots k_2(1,T) \\ & \vdots & \\ \frac{1}{2}k_1(T) & k_2(T,0) & k_2(T,1) \ldots k_2(T,T) \end{vmatrix} \tag{17}$$

The PDM were then defined as the eigenvectors with the largest eigenvalues (*Marmarelis, 1997*). The eigenvectors with the three largest (positive) eigenvalues were compared to $f_{linear}$ and $f_{nonlinear}$ by computing the cosine similarity according to *Equation 36*. The best matches were reported in all plots in Figures *Figure 3* and *Figure 3—figure supplement 1*. We note that for MINE, it was sufficient to compare the eigenvectors with the largest two eigenvalues; however, in the case of directly fitting the Volterra kernels (see below), the eigenvectors with the largest two eigenvalues often included an eigenvector related to the constant term $k_0$.

As a comparison to MINE, a system identification approach, directly fitting the Volterra kernels, was followed. We note that by creating an appropriate design matrix $X$, the elements of $k_0$, $k_1$, and $k_2$ can be directly obtained through a linear regression fit:

$$X = \begin{vmatrix} 1 & s(0) & s(0-1) \ldots s(0-T) & s(0)s(0) & s(0)s(0-1) \ldots s(0-T)s(0-T) \\ 1 & s(1) & s(1-1) \ldots s(1-T) & s(1)s(1) & s(1)s(1-1) \ldots s(1-T)s(1-T) \\ & & \vdots & & \\ 1 & s(n) & s(n-1) \ldots s(n-T) & s(n)s(n) & s(n)s(n-1) \ldots s(n-T)s(n-T) \end{vmatrix} \tag{18}$$

$$r = X\beta \tag{19}$$

$$\Rightarrow \beta = \begin{vmatrix} k_0 \\ k_1(0) \\ \vdots \\ k_1(T) \\ k_2(0,0) \\ k_2(0,1) \\ \vdots \\ k_2(T,T) \end{vmatrix} \tag{20}$$

A linear regression fit was used accordingly, either using ordinary least squares or Ridge regression with varying penalties $\alpha$ as indicated in *Figure 3* and *Figure 3—figure supplement 1*. We note that the direct fits were also performed after pre-whitening the design matrix as this might help with model fitting. However, this did not improve median filter quality but increased the variance across simulations (data not shown).

Effective degrees of freedom were calculated/estimated according to *Hastie et al., 2009*. Specifically, for Ridge regression models, the effective degrees of freedom were calculated analytically according to

$$df(\alpha) = tr\left(X(X^T X + \alpha \mathbf{I})^{-1} X^T\right) \tag{21}$$

where $\alpha$ is the Ridge penalty, $\mathbf{I}$ is the identity matrix, and $tr$ is the trace operator.

For the CNN used in MINE, the effective degrees of freedom were estimated using simulations according to

$$df(\hat{y}) = \frac{1}{\sigma^2} \sum_{i=1}^{N} Cov(\hat{y}_i, y_i) \tag{22}$$

where $\hat{y}$ is the model prediction, $y$ is the true output, $\sigma^2$ is the error variance, and $Cov$ is the covariance. Intuitively, this definition of degrees of freedom measures how small variations in the output affect variations in the predicted output. In other words, models with high degrees of freedom are expected to match predictions to any change in output while models with low degrees of freedom are limited in this regard.

All fits were repeated for 100 randomly generated stimuli to obtain different estimates for both MINE and Volterra analysis.

*Related Python file in the repository: cnn_sta_test.py, mine_edf.py.*

## Identifying contributing predictors by Taylor decomposition

To assess the contribution of individual predictors on the network output, a metric based on the Taylor decomposition of the network was computed. The Taylor decomposition was truncated after the second derivative. Below, Taylor decompositions that include all terms for first- and second-order derivatives will be designated as 'full.' For all analyses presented in the article, Taylor expansions were performed every second (every five timepoints at our data rate of 5 Hz) and predictions were made 5 s into the future (25 timepoints at our data rate of 5 Hz).

The state of the predictors at timepoint $t$ and the state of the predictors 5 s (in our case, 25 samples) later was obtained. Subsequently, these quantities, together with the Jacobian at timepoint $t$ and the Hessian at timepoint $t$, were used to compute a prediction of the change in network output over 5 s. At the same time, the true change in output was computed. Comparing the predicted changes and true changes via correlation resulted in an $r^2$ value. This value measures how well the full Taylor expansion approximates the change in network output across experimental time. We note that predicted changes in the output were compared via correlation rather than predicted outputs since we specifically wanted to understand which predictors are important in driving a change in output rather than identifying those that the network might use to set some baseline value of the output. In the following, $X(t)$ is a vector that contains the values of all predictors across all timepoints fed into the network (e.g. in our case 50 timepoints of history for each predictor) in order to model the activity at time $t$, $f(x)$ is the CNN applied to a specific predictor input, $J(x)$ is the Jacobian at that input, and $H(x)$ the Hessian:

$$\vec{x} = X(t)$$
$$\vec{x'} = X(t + 25)$$

The Taylor expansion around $\vec{x}$:

$$\hat{f}(\vec{x'}) = f(\vec{x}) + (\vec{x'} - \vec{x})^T \vec{J}(\vec{x}) + \tfrac{1}{2}(\vec{x'} - \vec{x})^T \mathbf{H}(\vec{x})(\vec{x'} - \vec{x}) \tag{23}$$

The predicted change in network output according to the Taylor expansion:

$$\Delta\hat{f} = \hat{f}(\vec{x'}) - f(\vec{x}) \tag{24}$$

The true change in network output:

$$\Delta f = f(\vec{x'}) - f(\vec{x}) \tag{25}$$

Variance of the output change explained by the Taylor prediction using *Equation 24* and *Equation 25*:

$$r^2_{Full} = corr(\Delta f, \Delta\hat{f})^2 \tag{26}$$

After the computation of $r^2_{full}$, terms were removed from the Taylor series. We note that both individual timepoints fed into the network and predictors are separated at this level. However, only the removal of predictors onto the quality of prediction was tested, not the importance of specific timepoints across predictors (i.e. all timepoints across a given predictor or interaction were removed to test the importance of the predictors, instead of removing all predictors at one timepoint to test the importance of a specific timepoint). For the removal of a given predictor, both its linear (via $J$) and its squared contribution (via $H$) were subtracted from the full Taylor prediction while for the removal of interaction terms only the corresponding interaction term (via $H$) was subtracted. The following

quantities were computed to arrive at the Taylor metric of single predictors $x_n$ or interactions of two predictors $x_{n,m}$, where $x_n$ and $J_n$ identify subvectors that contain all elements belonging to the specific predictor $n$ while $H_{n,m}$ identifies all elements in the Hessian at the intersection of all rows belonging to the second derivatives with respect to predictor $n$ and columns with respect to the predictor $m$: change predicted by only considering $x_n$:

$$\Delta \hat{f}_{x_n} = (\vec{x_n^J} - \vec{x_n})^T J_n + \tfrac{1}{2}(\vec{x_n^J} - \vec{x_n})^T \mathbf{H_{n,n}}(\vec{x_n^J} - \vec{x_n}) \tag{27}$$

Change predicted when removing $x_n$ using *Equation 24* and *Equation 27*:

$$\Delta \hat{f}_{-x_n} = \Delta \hat{f} - \Delta \hat{f}_{x_n} \tag{28}$$

Change predicted when only considering interaction between $x_n$ and $x_m$:

$$\Delta \hat{f}_{x_n, x_m} = \tfrac{1}{2}(\vec{x_n^J} - \vec{x_n})^T \mathbf{H_{n,m}}(\vec{x_m^J} - \vec{x_m}) + \tfrac{1}{2}(\vec{x_m^J} - \vec{x_m})^T \mathbf{H_{m,n}}(\vec{x_n^J} - \vec{x_n}) \tag{29}$$

Change predicted when removing $x_n x_m$ interaction using *Equation 24* and *Equation 29*:

$$\Delta \hat{f}_{-x_n, x_m} = \Delta \hat{f} - \Delta \hat{f}_{x_n, x_m} \tag{30}$$

Variance explained by Taylor prediction after removing $x_n$ using *Equation 25* and *Equation 28*:

$$r_{xn}^2 = corr(\Delta f, \Delta \hat{f}_{-x_n})^2 \tag{31}$$

Variance explained by Taylor prediction after removing $x_n x_m$ interaction using *Equation 25* and *Equation 30*:

$$r_{xn,xm}^2 = corr(\Delta f, \Delta \hat{f}_{-x_n, x_m})^2 \tag{32}$$

The Taylor metric was then defined as

$$T_{x_n} = 1 - r_{xn}^2 / r_{Full}^2 \tag{33}$$

and

$$T_{x_n, x_m} = 1 - r_{xn,xm}^2 / r_{Full}^2 \tag{34}$$

respectively. For the zebrafish data, we additionally calculated an overall 'Behavior' Taylor metric in which all terms (regular and interaction) that belonged to any behavioral predictor were removed and the Taylor metric was subsequently calculated as outlined above. *Related Python file in the repository: taylorDecomp.py.*

## Clustering of neurons according to receptive field

Jacobians were extracted for each neuron at the data average and used as proxies of the receptive field (see section on 'Comparison to Volterra analysis/determination of receptive fields'). Accordingly, the receptive field of a network and therefore neuron ($RF_f$) was defined as

$$RF_f = J(\bar{x}) \tag{35}$$

This is of course merely an approximation for nonlinear neurons. The first-order derivative of a nonlinear function is not constant and therefore the receptive field will depend on the inputs. To cluster zebrafish sensory neurons according to the receptive fields, the cosine of the angle between the receptive fields of different neurons ($RF_i$, $RF_j$, $i \neq j$) (and during the process, clusters) was computed (cosine similarity, $CS$):

$$CS_{RF_i, RF_j} = \frac{RF_i^T RF_j}{\|RF_i\|_2 \|RF_j\|_2} \tag{36}$$

Clustering was subsequently performed by a greedy approach as outlined in *Bianco and Engert, 2015*. Briefly: all pairwise CS values were computed. Then, the pair of receptive fields with the highest CS was aggregated into a cluster. The average RF in that cluster was computed. Subsequently all pairwise CS values, now containing that new cluster average, were computed again and the procedure was

iteratively repeated until all receptive fields were clustered or no pairwise similarities $CS_{RF_i,RF_j} \geq 0.8$ were observed. The advantages of this clustering method are that it is (1) deterministic and (2) the number of clusters does not have to be specified beforehand. In the case of the receptive fields on the mouse dataset, the matrix of pairwise cosine similarities was used to perform spectral clustering. Spectral clustering finds groups of strong connection (high CS in this case) within a similarity graph that are separated from other groups by weak connections (low CS). The rationale for changing clustering methods was computational speed, which is much higher for spectral clustering.

*Related Python file in the repository: utilities.py.*

## Ground-truth datasets

To test the capability of MINE to predict the relationships between predictors and neural activity, four predictors were computer-generated. Two of these were generated by combining sine, square, and triangle waves of random frequencies and amplitudes – simulating smoothly varying sensory inputs. The third predictor was generated as a Poisson process – simulating stochastic, unitary events such as bouts of movement. The last predictor was generated as a Poisson process as well but events were scaled with a random number drawn from a unit normal distribution with mean zero – simulating directed, stochastic movements.

Model responses were subsequently created by linearly or nonlinearly transforming individual predictors and combining them multiplicatively. After this step, all responses were convolved with a 'calcium kernel' and *i.i.d.* Gaussian noise was added to more closely resemble real-world acquisition. MINE was applied to these data: fitting a CNN on two-thirds of the data and calculating the correlation between the CNN prediction and the last third (test set) of the data. As a comparison, two linear regression models were fit to the data as well, again split into training and test sets. One of these models used raw predictors as inputs to predict the response (equivalent to the inputs given to MINE). The other used transformed predictors, convolved with the same calcium kernel applied to the responses above, along with all first-order interaction terms of the predictors to predict the response. Correlations on the test data were subsequently used for comparisons.

*Related Python file in the repository: cnn_fit_test.py.*

To test the capability of MINE in predicting the nonlinearity of predictor–response relationships, for each test case a new 'sensory-like predictor' (introduced above) was generated. For the effect of varying nonlinearity (*Figure 3B–D*), the predictor was transformed by (1) squaring the hyperbolic tangent of the predictor, (2) differencing the predictor, or (3) rectifying the predictor using a 'softplus' transformation. These transformed versions were subsequently mixed to varying fractions with the original predictor to generate a response. The coefficients of determination of truncations of the Taylor expansion after the linear and quadratic terms were then computed as indicated above. Linear correlations were also performed for some of the mixtures to illustrate the effect of both linear and nonlinear transformations on this metric. For ROC analysis of the linearity metric, a set of 500 by 5 random stimuli were generated. For each set, four responses were generated depending only on the first of the five random stimuli (the others were used as noisy detractors to challenge the CNN fit). Two of the four responses were linear transformations: one was a differencing of the first predictor and the other was a convolution with a randomly generated double-exponential filter. The other two responses were randomized nonlinear transformations. The first was a variable-strength rectification by shifting and scaling a softplus function. The second was a random integer power between one and five of a hyperbolic tangent transformation of the predictor. The known category (linear vs. nonlinear) was subsequently used to calculate false- and true-positive rates at various thresholds for the linearity metric (see above) to perform ROC analysis.

*Related Python file in the repository: cnn_nonlin_test.py.*

## Processing of data from *Musall et al., 2019b*

The data for Musall (*Musall et al., 2019a*) was downloaded from the published dataset. Temporal components *Musall et al., 2019a* were loaded from 'interpVc.mat,' spatial components from 'Vc. mat,' and predictors from 'orgregData.mat' for each analyzed session. Predictors were shifted by 5 s relative to the activity traces. The 10-s-long convolutional filters allowed for observing the influences of events 5 s into the past as well as preparatory activity for actions occurring in the next 5 s. MINE was subsequently used to determine the relationship between predictors and temporal components. The

neural activity and the individual terms of the Taylor decomposition were mapped into pixel space. Then spatial maps were generated by computing Taylor metrics as above for each individual pixel time series. This was necessary since the components in 'interpVc.mat' are highly linearly dependent, making it impossible to carry measures of explained variance directly from 'component space' into 'pixel space.' Receptive fields, on the other hand, were directly mapped into pixel space using the spatial component matrix as these are linear. *Related Python files in the repository: processMusall.py, mine.py, plotMusall.py.*

## Zebrafish experiments

All calcium imaging was performed on a custom-built two-photon microscope at 2.5 Hz with a pixel resolution of 0.78 *μm/pixel* and an average power of 12 mW (at sample) at 950 nm (measured optical resolution of the system is $< 1\ \mu m$ lateral and $< 4\ \mu m$ axial). The brain was divided into six overlapping regions: two divisions along the dorsoventral axis and three divisions along the anterior–posterior axis. The most anterior point in the most anterior region was set to the olfactory epithelium while the most posterior point in the most posterior region was the start of the spinal cord. Larval zebrafish expressing nuclear localized GCaMP in all neurons and mCherry in all glutamatergic neurons (mitfa -/-; elavl3-H2B:GCaMP6s +/-; vGlut2a-mCherry) between 5 and 7 d post fertilization were used for all experiments. Larval zebrafish were mounted and stimulated as previously described (*Haesemeyer et al., 2018*) including scan stabilization as previously described to avoid artifacts associated with heating-induced expansion of the preparation. The tail of the larva was free and tracked at 250 Hz using an online tracking method previously described (*Haesemeyer et al., 2018*). Whenever our online tail-tracking missed a point along the tail, an image was saved from which the tail was subsequently retracked offline as in *Mearns et al., 2020*. Since the experiments were performed under a new microscope, we recalculated a model translating stimulus laser strength to fish temperature as described in *Haesemeyer et al., 2015*; *Haesemeyer et al., 2018* and subsequently referred to as the 'temperature model'.

For each imaging plane, 495 s of a 'random wave' stimulus were generated by superimposing sine waves of randomized amplitudes with periods between 13 s and 200 s. The stimuli were set to generate temperatures outside the noxious range between 22°C and 30°C. Imaging planes were spaced 5 *μm* apart and 30 planes were imaged in each fish.

For reticulospinal backfills Texas-Red Dextran 10,000 MW (Invitrogen D1828) at 12.5% w/v was pressure injected into the spinal cord of larval zebrafish under Tricaine (Syndel MS222) anesthesia. Injections were performed in fish expressing nuclear localized GCaMP6s in all neurons (mitfa -/-; Elavl3-H2B:GCaMP6s) that were embedded in low-melting point agarose with a glass capillary having a 15 *μm* tip opening. Fish were unembedded after the procedure, woken from anesthesia, and rested overnight. Fish were screened the next day for labeling, embedded again, and imaged under the two-photon microscope.

## Processing of zebrafish data

Raw imaging data was preprocessed using CaImAn for motion correction and to identify units and extract associated calcium traces (*Giovannucci et al., 2019*). Laser stimulus data was recorded at 20 Hz and converted to temperatures using the temperature model (see above). The 250 Hz tail data was used to extract boutstarts based on the absolute angular derivative of the tail crossing an empirically determined threshold. The vigor was calculated as the sum of the absolute angular derivative across the swimbout, which is a metric of the energy of tail movement. The directionality was calculated as the sum of the angles of the tip of the tail across the swimbout, which is a metric of how one-sided the tail movement is. Extracted calcium data, stimulus temperatures, and the three behavior metrics were subsequently interpolated/downsampled to a common 5 Hz timebase. To reduce linear dependence, the behavior metrics were subsequently orthogonalized using a modified Gram–Schmidt process.

Components outside the brain as well as very dim components inside the brain were identified by CaImAn. To avoid analyzing components that likely do not correspond to neurons labeled by GCaMP, all components for which the average brightness across imaging time was <0.1 photons were excluded. The number of analyzed components was thereby reduced from 706,543 to 432,882.

The CNN was subsequently fit to the remaining components using two-thirds of imaging time as training data. For each neuron, the stimulus presented during recording and the observed behavioral

metrics were used as inputs. Both the inputs and the calcium activity traces were standardized to zero mean and unit standard deviation before fitting. For every neuron in which the correlation $r \geq \sqrt{0.5}$, all discussed MINE metrics were extracted.

To generate the cyclically permuted controls, all calcium data was rotated forward by a third of the experiment length with wrap around. The rationale for constructing the control data in this manner was that it maintains the overall structure of the data, but since both our stimuli and the elicited behavior are stochastic, it should remove any true relationship between predictors and responses. The CNN was fit to the control data in the same manner as to the real data, but no further metrics were extracted. To identify which predictors significantly contributed to neural activity, $R_{full}$, $R_{Pn}$, and $R_{Pn,m}$ were bootstrapped, computing the Taylor metric for each variate. The standard deviation of the bootstrap variate was subsequently used to estimate confidence intervals according to a normal distribution approximation. While confidence intervals could have been computed directly from the bootstrap variates, this would have meant storing all samples for all neurons in questions to retain flexibility in significance cutoff. The significance level was set to p<0.05 after Bonferroni correction across all fit neurons (effective p-value of $1.26 \times 10^{-6}$). To be considered a driver of neural activity, the Taylor metric had to be larger than 0.1 at this significance threshold.

The linear comparison model was constructed akin to *Musall et al., 2019a*. All predictors were timeshifted by between 0 and 50 timesteps for a total of 200 predictor inputs to the regression model. This setup emulates the convolutional filters present in the CNN used by MINE. A modified Gram–Schmidt process was used for orthogonalization to avoid the problems of QR decomposition observed in cases of near singular design matrices (singular up to the used floating-point precision). Ridge regression (*Hastie et al., 2009*) was used to improve generalization. Setting the ridge penalty to $10^{-4}$ led to a 25–50% increase of identified neurons on small test datasets compared to a standard regression model.

All zebrafish stacks (except those acquired after spinal backfills) were first registered to an internal Elavl3-H2B:GCaMP6s reference using CMTK (*Rohlfing and Maurer, 2003*). A transformation from the internal reference to the Z-Brain (*Randlett et al., 2015*) was subsequently performed using ANTS (*Avants et al., 2009*). Micrometer coordinates transformed into Z-Brain pixel coordinates, together with region masks present in the Z-Brain, were subsequently used to assign all neurons to brain regions.

To identify Z-Brain regions in which a specific functional type is enriched, the following process was used. The total number of all types under consideration in each region was computed (i.e. if the comparison was between stimulus, behavior, and mixed-selectivity neurons, all these neurons were summed, but if the comparison was between different classes of behavior-related neurons only those were summed). Subsequently, the fraction of the type of interest among the total in each region was computed (plotted as bars in *Figure 5—figure supplement 1* and *Figure 6—figure supplement 1*). Similarly, an overall fraction of the type across the whole brain was computed (plotted as vertical blue lines in *Figure 5—figure supplement 1* and *Figure 6—figure supplement 1*) This overall fraction was used to simulate a 95% confidence interval of observed fractions given the total number of neurons considered in each region (horizontal blue lines in *Figure 5—figure supplement 1* and *Figure 6—figure supplement 1*). Any true fraction above this interval was considered significant enrichment. We note that exact confidence intervals could have been computed from the binomial distribution; however, the simulation approach allowed us to use the same method when computing confidence intervals around the average complexity of receptive field clusters.

For reticulospinal backfill experiments, neurons labeled by the backfill were manually annotated. This annotation was used to classify neurons (CaImAn identified components) as reticulospinal.

To study how mixed-selectivity neurons integrate information about the temperature stimulus and swim starts, the following strategy was used. Receptive fields for the temperature and swim start predictor were extracted for each neuron as described above. Subsequently, the drive for an input was defined as a scaled version of the receptive fields after they had been scaled to unit norm. Synthetic input stimuli at different combinations of stimulus and swim drive were subsequently generated and fed into the CNN of the neuron of interest. The response of the CNN was recorded to construct the response landscapes shown in *Figure 7* and *Figure 7—figure supplement 1*. To cluster similar response landscapes, pairwise 2D cross-correlations were computed between the response landscapes of all neurons. Similarity was defined as the maximal value of the cross-correlation to preferentially cluster on similar shapes of the landscape rather than exact alignment. Spectral clustering

was used to form the clusters and for each a randomly selected exemplar is shown in the figures. Due to the lack of alignment, cluster averages are not meaningful.

To perform anatomical clustering, every identified neuron was clustered based on the spatial proximity of their centroids to the centroids of other neurons. A radius of 5 $\mu m$ (roughly the radius of a neuron in the zebrafish brain) was chosen empirically as the clustering radius. The coordinates of the centroids were then placed in a 3D volume with a spatial resolution of 1 $\mu m$, with each centroid occupying one position in this space. Subsequently, a kernel of the chosen radius was constructed. The kernel was then convolved with the centroids of the neurons to generate a ball around each of them. The goal was that if the space occupied by the balls around two neurons overlap, or are adjacent, the two neurons would be clustered together. This was accomplished using connected-components-3D (William Silversmith, version 3.12, available on PyPi as connected-components-3d; code: https://github.com/seung-lab/connected-components-3d/, copy archived at *Silversmith, 2023*) which was used to cluster the neurons based on their proximity. The connected-components-3D function takes the 3D volume and assigns each occupied voxel to a cluster. The corresponding centroids were then identified and assigned the same cluster label. For plotting, an alpha value was assigned to each centroid based on the number of neurons present in each cluster. The coordinates of the centroids were then plotted and assigned the computed alpha values.

Experiments were excluded from further analysis for the following reasons: five fish died during functional imaging; eight fish could not be registered to the reference; tail tracking failed during the acquisition of six fish and during the acquisition of four fish problems in the acquisition hardware led to imaging artifacts. *Related Python files in the repository: rwave_data_fit.py, rwave_decompose.py, utilities.py.*

## Acknowledgements

We thank Andrew D Bolton and Bradley Cutler for critical comments on the manuscript and the 'Zebrafish Neural Circuits & Behavior' seminar community for early stage feedback and discussion. We would also like to thank participants of NeuroDataShare 2023 for helpful discussions about this work and suggested improvements to our sharing of data. This work was supported by funding from the Ohio State University and NIH BRAIN Initiative R01-NS123887 through NINDS and NIDA to MH.

## Additional information

### Competing interests
Jamie D Costabile: is employed by Hitachi Solutions America, Ltd., Irvine, CA. The other authors declare that no competing interests exist.

### Funding

| Funder | Grant reference number | Author |
| --- | --- | --- |
| National Institutes of Health | 5R01NS123887-02 | Jamie D Costabile<br>Kaarthik A Balakrishnan<br>Martin Haesemeyer |
| The Ohio State University Wexner Medical Center | | Sina Schwinn<br>Martin Haesemeyer |

The funders had no role in study design, data collection and interpretation, or the decision to submit the work for publication.

### Author contributions
Jamie D Costabile, Conceptualization, Data curation, Software, Formal analysis, Methodology, Writing – review and editing; Kaarthik A Balakrishnan, Formal analysis, Investigation, Methodology, Writing – review and editing; Sina Schwinn, Investigation; Martin Haesemeyer, Conceptualization, Software, Formal analysis, Supervision, Funding acquisition, Visualization, Methodology, Writing - original draft, Project administration, Writing – review and editing

**Author ORCIDs**
Martin Haesemeyer ⬤ http://orcid.org/0000-0003-2704-3601

## Ethics

Animal handling and experimental procedures were approved by the Ohio State University Institutional Animal Care and Use Committee (IACUC Protocol #: 2019A00000137 and 2019A00000137-R1).

## Decision letter and Author response

Decision letter https://doi.org/10.7554/eLife.83289.sa1
Author response https://doi.org/10.7554/eLife.83289.sa2

# Additional files

## Supplementary files

- Supplementary file 1. Z-Brain region abbreviations.
- MDAR checklist

## Data availability

All data generated in this study is publicly available. All code used in this study is available in the following repositories: MINE: https://github.com/haesemeyer/mine_pub (copy archieved at *Haese-meyer, 2023*). CalmAn preprocessing: https://github.com/haesemeyer/imaging_pipeline (copy archieved at *Haesemeyer, 2021*). Tail behavior processing: https://bitbucket.org/jcostabile/2p/. Raw experimental data has been deposited to DANDI: https://dandiarchive.org/dandiset/000235/0.230316.1600; https://dandiarchive.org/dandiset/000236/0.230316.2031; https://dandiarchive.org/dandiset/000237/0.230316.1655; https://dandiarchive.org/dandiset/000238/0.230316.1519. Processed data has been deposited on Zenodo: Zebrafish and mouse final processed data: https://doi.org/10.5281/zenodo.7737788. Zebrafish and mouse fit CNN weights part 1/2: https://doi.org/10.5281/zenodo.7738603. Zebrafish and mouse fit CNN weights part 2/2: https://doi.org/10.5281/zenodo.7741542. All data analysis was performed in Python; Tensorflow (*Abadi, 2016*) was used as the deep learning platform.

The following datasets were generated:

| Author(s) | Year | Dataset title | Dataset URL | Database and Identifier |
|---|---|---|---|---|
| Balakrishnan KA, Schwinn S, Costabile JD, Haesemeyer M | 2023 | Processed data for "Model-free identification of neural encoding (MINE)" publication | https://doi.org/10.5281/zenodo.7737788 | Zenodo, 10.5281/zenodo.7737788 |
| Balakrishnan KA, Haesemeyer M | 2022 | Thermoregulatory Responses Forebrain | https://doi.org/10.48324/dandi.000235/0.230316.1600 | Dandi Archive, 10.48324/dandi.000235/0.230316.1600 |
| Balakrishnan KA, Haesemeyer M | 2022 | Thermoregulatory Responses Midbrain | https://doi.org/10.48324/dandi.000236/0.230316.2031 | Dandi Archive, 10.48324/dandi.000236/0.230316.2031 |
| Balakrishnan KA, Haesemeyer M | 2022 | Thermoregulatory Responses Hindbrain | https://doi.org/10.48324/dandi.000237/0.230316.1655 | Dandi Archive, 10.48324/dandi.000237/0.230316.1655 |
| Schwinn S, Haesemeyer M | 2022 | Thermoregulatory Responses Reticulospinal system | https://doi.org/10.48324/dandi.000238/0.230316.1519 | Dandi Archive, 10.48324/dandi.000238/0.230316.1519 |
| Balakrishnan KA, Schwinn S, Costabile JD, Haesemeyer M | 2023 | CNN weight data for "Model-free identification of neural encoding (MINE)" publication - Set 1 | https://doi.org/10.5281/zenodo.7738603 | Zenodo, 10.5281/zenodo.7738603 |
| Balakrishnan KA, Schwinn S, Costabile JD, Haesemeyer M | 2023 | CNN weight data for "Model-free identification of neural encoding (MINE)" publication - Set 2 | https://doi.org/10.5281/zenodo.7741542 | Zenodo, 10.5281/zenodo.7741542 |

The following previously published dataset was used:

| Author(s) | Year | Dataset title | Dataset URL | Database and Identifier |
|---|---|---|---|---|
| Churchland AK, Musall S, Kaufman MT, Juavinett AL, Gluf S | 2019 | Dataset from: Single-trial neural dynamics are dominated by richly varied movements | https://doi.org/10.14224/1.38599 | CSHL, 10.14224/1.38599 |

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
