## [Editor Report]

This useful article describes a sensitive method for identifying the contributions of different behavioral and stimulus parameters to neural activity. The method has been convincingly validated using simulated data and applied to example state-of-the-art datasets from mouse and zebrafish. The method could be productively applied to a wide range of experiments in behavioral and systems neuroscience.

---

## [Decision Letter]

**Decision letter after peer review:**

Thank you for submitting your article "A model-free approach to link neural activity to behavioral tasks" for consideration by *eLife*. Your article has been reviewed by 3 peer reviewers, and the evaluation has been overseen by a Reviewing Editor and Michael Frank as the Senior Editor. The reviewers have opted to remain anonymous.

Essential revisions:

The reviewers were positive about the paper's approach. Below are several points agreed upon by the reviewers that would be important to address before potential publication.

1) R1 had several critiques of the comparison to STA/STC. All reviewers agree that a more complete exposition of the differences between this method and STA/STC would benefit the paper. This could include synthetic cases where the proposed methods works better than STA/STC (for instance requiring less data) or real examples from the datasets where the proposed method is better at classifying cell response types. This paper should describe or show the advantages of this type of analysis over prior ones, ideally using good apples-to-apples comparisons.

a. Relatedly, R1 was also skeptical of the utility of the linear methods shown in Figure 1; the other reviewers convinced R1 that the limited linear models used were in keeping with standard methods in the zebrafish field. The authors should better describe why these particular (limited) linear models were used, and perhaps cite examples from the field in support of these sorts of models.

2) The paper was a bit dense in some sections. The reviewers recommend moving some more mathematical details to the supplement and focusing on the most important results in the main text. (An example might be the curvature metric, which was a little opaque and did not seem central.)

3) The authors should draw finer distinctions between their method and some other, similar, prior methods, like the Ganguli/Baccus papers that used convolutional networks to fit retinal ganglion cell responses, in an approach that seems similar in some ways to the one used here. This relates also to several reviewer comments about what method exactly is being referred to as 'MINE'.

4) Figure 7: The authors should explain the significance of the result in Figure 7 with respect to signal processing. Are the authors saying that higher order interactions occur deeper in processing? This complexity analysis may also be an example of an approach that is possible with what is presented here but not with STA/STC.

*Reviewer #1 (Recommendations for the authors):*

1) In MINE's first mention (line 43) and in showing that it can capture various response features, the authors seem to imply that MINE is simply the idea of using ANNs to model neural activity as function of either independent or dependent variables. But surely this has already been done many times. I'm having some trouble seeing what precisely is claimed as new in MINE. My thought was that it is also the first and second order interpretation, but that does not come through in the early descriptions of the method.

2) Related to this, in L55, the authors remark "Aside from network architecture and initialization conditions MINE is essentially model-free." This is a pretty big constraint, by the model architecture, choice of activation function, etc. STA and STC and systems identification methods more generally seem far more truly model free. This suggests that the framing and title here might need significant reconsideration.

3) The authors make several claims about linear models in Figure 1 that I cannot figure out. In particular, at L137, the authors say that the linear model cannot learn the dynamics of the calcium indicator. But why not? If the stimulus is s_t, the kernel is k_t, and the response is r_t = (s*k)_t + eta_t, then this is exactly what a linear model should be able to learn perfectly, directly from the data through a least squares fit at each time point. I don't understand why an expanded linear model should be required to learn the calcium kernel. In 1F, why is the LM not working as well as the ANN for S1 and M2? Those appear to both be simple linear transformations where r_t = (s*k)_t + eta_t: perfect for a linear model and least squares fitting. (In fact, with Gaussian noise, no model should be able to outperform the least squares fit linear model to estimate k, given sufficient data.) Similarly, dS1/dt is still just a linear transformation of S1 and should be easily learned by a linear model, since it's now just r_t = (s*k')_t + eta_t, where k' is the derivative of the calcium kernel. These results – that the authors' linear model is not capturing linear transformations that exactly match the form of the linear model – are truly puzzling.

4) One of the big proposed advances of this paper is the interpretability of the fitted ANN model. By doing "Taylor expansion" on the model, the authors pull out linear and quadratic terms that describe the dependency of neurons on different non-neural variables (dependent and independent) in these experiments. I have a few comments on this, some more critical than others. Overall, I was not left with a strong sense of what the benefit is, if any, of this method over standard systems identification approaches.

a. This sort of expansion of a functional has more typically been called a Volterra expansion, and there exists a giant literature on estimating Volterra kernels for nonlinear systems. It would be appropriate to cite some of that work and adopt the terminology used in prior work in the field.

b. In comparing their methods to spike-triggered analyses, Figure 4 is not a fair comparison, since it has not decorrelated the stimuli in time, as has been in wide practice in systems identification for quite a while (see Korenberg in the 1990s, Baccus and Meister, 2002). Thus the fair comparison is with S4, and 4 should not be viewed as comparable and quite likely removed.

c. In the comparison between the Jacobian/Hessian of the ANN and the STA/STC equivalents in Figures4 and S4, I don't understand the authors' choice of nonlinearities, which are described after line 1105. In particular, g_sta(x) has a first derivative at x=0, but it's second derivative evaluated at x=0 is also non-zero. By adding this to the other function and putting them through a ReLU (equation 19), there will be both linear and quadratic components for both RFa and RFv, even if g_stc is symmetric, simply because the ReLU is approximated by a linear + quadratic (+ higher) terms. This means that with these functions, I would expect the STA to be a linear combination of the two linear projections (RFa and RFv) and I would expect the STC matrix to have eigenvectors spanning the 2-dimensional space of RFa and RFv. It's less clear why this isn't happening in the simulations. RFa could dominate the STA. In Figure 4B, it seems clear that the STC is contaminated by the STA, and given the analysis above, that is not surprising. Is that what is lowering the cosine similarity for the STC analysis in S4? Is only the first eigenvector being considered? If so, it's frequently the case that the STC dimensions also include the STA dimension; often, one reports only the STC components after the STA has been subtracted off. (See for instance Aljadeff et al., 2016.) It looks here as though the authors just report the eigenvector with largest eigenvalue. Is the STC really not working well, or is it just including components of the other filter (appropriately)? It's hard to believe that STC applied to a point nonlinearity acting on two projections of the stimulus would not return eigenvectors that span those two projections of the stimulus.

d. This brings up a conceptual point. The decomposition that the authors perform of the ANN into zeroth, first, second, and higher order terms in Figure 7D could just as easily be written for the raw responses of the neurons. In such a case, with an excellent model, the two expansions should be closely related to one another, and in some cases equivalent. The low-order terms in the Volterra series, when estimated experimentally, depend on higher order terms. (For instance a cubic Volterra term would show up in the linear filter.) Wiener series are typically estimated instead, since these produce an orthogonalized basis, so that this dependence is eliminated (for a given amplitude of inputs X). When computing the STC, one can eliminate the linear component first; that would then be the Wiener estimate of the second order kernel. The second order kernel computed as the Hessian of an ANN is not orthogonalized in the same way, so it might produce different results from STC, but it is not clear that this is what the authors show, or intend to show.

e. The authors on L346 say that higher order terms are contained in the ANN, so that the Jacobian will be different from spike triggered covariance analysis. I think this is correct. However, I'm not sure that makes the Hessian or Jacobian in the ANN a better analysis than the systems identification approaches. In my mind, this is potentially the only advantage of doing this ANN method over STA and STC; but if that's the advantage, then I'd want to be shown very clearly an example where this occurs and where the insight into the system was improved by using the Jacobian/Hessian rather than STA/STC. I imagine that such benefits could accrue when the fitted ANN is close to the true set of transformations of the system, but that would reduce the advantages of being able to do this with any old ANN. Overall, these comments get to the point of: under what conditions would the Jacobian/Hessian/ANN method be better or somehow more informative than STA/STC? And why fit to the ANN if one is going to interpret it with methods equivalent to STA/STC that could be performed directly on the data?

*Reviewer #2 (Recommendations for the authors):*

While the paper could certainly be published in the current form, there are some points that as a reader I would like to have seen addressed further. I therefore offer some comments and suggestions that the authors might like to consider.

I believe there is a mistake in the formulae given for the Taylor metric in the Methods. Since the R^2 Full is always greater than the R^2 with fewer parameters, the formulae, as written, appear to give a number between 0 and -infinity, whereas I think the intention is that they should give a number between 0 and 1 (as appears in the paper).

An important consideration when comparing approaches to modeling neuronal responses is how much data is required to fit the model. In the comparison with regression models, or with the spike-triggered average it would be useful to show how the performance of each approach develops with the length of recording.

The paper could be improved, and the method made more appealing, by more clearly showing qualitative improvements that their approach offers over variations of linear regression methods. In the zebrafish part, there is a convincing argument made that the method allows more neurons to be fit at some threshold, but direct comparison of the outputs of both approaches is restricted to counting the numbers of ROIs identified in various classes. One possible explanation could be that these differences arise because the model-free method has greater sensitivity for identifying neurons within the same classes that are found with the LM method, but it would be more interesting if the model free method finds particular types of neurons that are clearly missed by the LM method. What could be compelling to see, for example, would be (1) differences in the spatial distributions of neuron classes found with the two methods, e.g. regions where neurons are only found with the model-free method or (2) Evidence that there are specific functional classes that are missed with the LM method – e.g. do the RF clusters vary in how well they are picked up by the LM?

It would be helpful to have some more information about the choice of network architecture, and how this was optimized. For example, 80 features in the convolutional layer seems like a lot for these relatively simple data sets, and it would be good to understand better how this number was chosen and how this choice can affect the outcome.

Intuitively, it seems like the temporal effects of the calcium indicator response would ideally be applied downstream of the computation of the neural response, but, in the text, this step is described as happening in the first layer of linear convolutional filters. Is this a constraint of the network design, and are there any practical consequences for the possible computations captured by the model?

I don't like that the distributions of neuronal types seem to be shown in the form of renderings of anatomical masks highlighting regions of enrichment. This rather obscures the true distributions, and makes them appear more cleanly separated than they are. For me, these visualizations don't offer much information that goes beyond the tabulation of regions. A direct representation of the density of different functional classes, as is often used in the field, would be richer in information, and might be a way to highlight qualitative differences between the populations identified using the LM and model-free approaches. Even if it is the case that there are not strong differences in the distributions of functional types, this by itself would be an interesting observation that merits discussion.

I would like to see more discussion of how the neural network models could be explored or leverage to give more insight into the responses of individual neurons. For example, having identified the relevant variables, and model complexity, could this information be used to explore simpler models in a more constrained way? Could visualization approaches like those used in Heras et al. 2019 https://doi.org/10.1371/journal.pcbi.1007354, be used to explore how the network is combining task variables?

Since the function is scalar valued, I believe that what is referred to in the paper as the Jacobian is also simply the gradient of the function. Perhaps this could be mentioned explicitly to aid understanding for a broad audience.

Overall, I congratulate the authors on developing this useful method, and I look forward to seeing it published.

*Reviewer #3 (Recommendations for the authors):*

1. Overall, as mentioned in my Public Review section, I believe the paper is very valuable and merits publication in *eLife*. I do think that it would be a pity if the manuscript was not carefully re-read and rewritten to make it more accessible for the average reader who may want to implement this analysis on their data. An option would be to shift more material to the Methods and use the main text to convey the method and compare it with the available ones for the 3 datasets that are treated.

2. It would help to understand the various thresholds that appear throughout. I am, for example, not sure what to conclude from the sentence that starts on line 224 or how to interpret the red dashed line in Figure 5B (line 421).

3. Given that the zebrafish experiments were performed by the authors, could they comment on the reliability (across animals) in finding the functional activity described in Figures 6 and 7?

4. The code and data availability is exemplary.

[Editors' note: further revisions were suggested prior to acceptance, as described below.]

Thank you for resubmitting your work entitled "A model-free approach to link neural activity to behavioral tasks" for further consideration by *eLife*. Your revised article has been evaluated by Michael Frank (Senior Editor) and a Reviewing Editor.

The manuscript has been much improved, but there is one large and a few small remaining issues that need to be addressed, as outlined below:

There is only one issue that remains, and it is the issue of calling this a "model-free" approach, noted by R1. Since no analysis is truly model free, there is a spectrum of how much constraint a model puts on an analysis. There are other approaches discussed in the paper that are closer to the model free end than this one (like STA, STC, MID), so this term does not seem like the right one to use in this case. The authors should find a more accurate description of what their method brings to field.

*Reviewer #1 (Recommendations for the authors):*

I appreciate the care that the authors have taken in addressing the concerns raised in the first round. I believe that the major hurdles were cleared in the revision. However, I do have a few comments that I believe should be addressed before final acceptance/publication.

1) This method is definitely not "model free". It is a relaxed model requirement, or potentially a very general model, or a model with few assumptions, but I don't think it's just a matter of opinion as to whether it's "model-free". Especially in comparison with STA and STC and Maximally informative dimensions, which all have even less of an underlying model framework… At a very basic level, the title and acronym must be accurate. Later in the paper, the authors argue that this method's model is actually providing important regularizing power in computing STC, etc. So it is definitely not model free, and the authors must find a different description for the title (and the acronym). I'm sure they can do this, but some suggestions: "Flexible INE"? CNNINE? "Canine"? "Convolutional neurAl Network Identification of Neural Encoding"? Something else? "Model-agnostic"?

2) Figure 7BC. The temporal kernels found by this method are a bit weird and seem not very biological – much of the power lies at 0 and 10 seconds, the two ends of the filter… Is this because the filters aren't long enough? These look quite different from true filters I'd expect to find in neurons, both in their duration and shape. I think this deserves some kind of explanation. I apologize that I didn't note this on the first round – I clearly got distracted by some other parts of the analysis.

---

## [Author Response]

Essential revisions:The reviewers were positive about the paper's approach. Below are several points agreed upon by the reviewers that would be important to address before potential publication.

We thank the reviewers for carefully engaging with our manuscript and providing constructive criticism. We believe that our revisions address the raised concerns and considerably strengthen the paper. We provide detailed responses to individual criticisms below. We have reorganized the paper and simplified our definition of nonlinearity with the intention to make the paper less dense and easier to follow. We have also rewritten our comparison to STA/STC, explicitly comparing MINE to Volterra analysis to demonstrate that MINE provides a simpler and more robust way to extract receptive fields. We specifically note that it wasn’t our intention to claim that MINE could do something that all well-crafted model-based methods fail at, but rather that MINE provides an accessible framework for performing crucial analyses on the relationship of predictors and neural activity. We specifically highlight these points in different parts of the paper.

1) R1 had several critiques of the comparison to STA/STC. All reviewers agree that a more complete exposition of the differences between this method and STA/STC would benefit the paper. This could include synthetic cases where the proposed methods works better than STA/STC (for instance requiring less data) or real examples from the datasets where the proposed method is better at classifying cell response types. This paper should describe or show the advantages of this type of analysis over prior ones, ideally using good apples-to-apples comparisons.

We have changed our strategy of comparing MINE to system-identification approaches. We specifically point out that because of the structure of the CNN underlying MINE (taking predictors across time as inputs) the Taylor expansion of the network is in fact equivalent to the Volterra expansion of a system with finite memory corresponding to the time-length of the inputs given to MINE (Lines 150-153).

We use synthetic data simulating the action of two different filters and nonlinearities on the data to compare MINE to directly fitting the Volterra kernels via linear regression, an approach that in theory is truly model-free. Just like MINE, ordinary linear regression recovers the filters without problem when the input is Gaussian white noise. This highlights the fact that the comparison is indeed fair: Given constraints on the stimulus, our simulation allows full recovery of the filters. Approaching more naturalistic inputs however, ordinary linear regression fails to recover Volterra kernels from which the filters can be extracted while the filters can still be derived using MINE. We then show that increasing an L2 penalty using Ridge regression allows us to indeed recover the filters from directly fitting the Volterra kernels. However, depending on the structure of the filter, MINE still recovers more faithful representations and MINE is overall better in generalizing to new data when compared with the best-fit Ridge regression models (Lines 169-183).

We furthermore show that in the process of increasing the Ridge penalty, the effective degrees of freedom of the Ridge regression model approach the effective degrees of freedom of MINE. In other words, system analysis can be indeed truly model-free, e.g. by imposing no constraints on a regression model that fits the Volterra kernels, however, such an approach does not yield useful results. To obtain useful results, the degrees of freedom of the model have to be reduced, either through penalties on a regression fit or by imposing a structured neural network (Lines 193-200). We would also like to point out that the reduction in degrees of freedom is a general theme when applying system analysis to cases in which inputs are not gaussian white noise, whether this is through the use of basis functions, artificial neural networks or constrained regression.

a. Relatedly, R1 was also skeptical of the utility of the linear methods shown in Figure 1; the other reviewers convinced R1 that the limited linear models used were in keeping with standard methods in the zebrafish field. The authors should better describe why these particular (limited) linear models were used, and perhaps cite examples from the field in support of these sorts of models.

We better clarify our choice of model in the first section of the paper. Convolving inputs with a presumed calcium kernel and subsequently generating a model that contains all possible interaction terms is indeed common and we provide corresponding citations from the zebrafish and mouse literature. We also specifically state in the text now (Lines 80-82), that it is not our goal to show that MINE can do something **no other model can** but rather that MINE has advantages over predefined models by flexibly learning data transformations. We would also like to point out that in the comparison to the zebrafish data we already use a regression model that can learn temporal transformations and in spite of that MINE identified more neurons.

2) The paper was a bit dense in some sections. The reviewers recommend moving some more mathematical details to the supplement and focusing on the most important results in the main text. (An example might be the curvature metric, which was a little opaque and did not seem central.)

We have attempted to make the paper less dense. We hope that the reordering of sections as well as the simplification of measuring nonlinearity help with this goal. After showing the capability of MINE to fit arbitrary data relationships we now introduce the Taylor expansion in a section that introduces the derivation of nonlinearity and complexity. Both of these metrics are now based on truncations of the Taylor series which removes the need to introduce curvature and the NLC in detail and which should also increase interpretability of our definition of nonlinearity. We subsequently use the same Taylor expansion to explain the extraction of receptive fields which we contrast with Volterra analysis. Subsequently we provide details for identifying contributing predictors, again via Taylor expansion but this time in a point-wise manner to account for nonlinear relationships, motivated by the section on nonlinearity and complexity. We hope that this flow will make it easier for readers to follow along.

3) The authors should draw finer distinctions between their method and some other, similar, prior methods, like the Ganguli/Baccus papers that used convolutional networks to fit retinal ganglion cell responses, in an approach that seems similar in some ways to the one used here. This relates also to several reviewer comments about what method exactly is being referred to as 'MINE'.

We have addressed this point. We specifically discuss the Ganguli/Baccus papers in the discussion (Lines 482-488). We believe that this discussion leads to new insight in relation to some of our discoveries in the comparison to Volterra analysis. We also fully agree that we never explicitly clarified what MINE is. We now specifically state in the introduction (Lines 21-26) and at the beginning of the discussion (Lines 446-452) that we understand MINE as the combination for obtaining a predictive model of neural activity giving predictor input, characterizing the nonlinearity/complexity of the relationship between inputs and neural activity, identifying receptive fields and characterizing which inputs drive neural activity. As such MINE provides an accessible and easy-to-use interface for a nearly model-free characterization of how neurons encode task-relevant variables.

4) Figure 7: The authors should explain the significance of the result in Figure 7 with respect to signal processing. Are the authors saying that higher order interactions occur deeper in processing? This complexity analysis may also be an example of an approach that is possible with what is presented here but not with STA/STC.

We now explicitly discuss the significance of mapping computational complexity for understanding functional neural circuits (Lines 525-530). We have also increased our analysis of mixed-selectivity neurons highlighting another feature of MINE: Using the power of the predictive model to better understand how neurons integrate different features in our case thermosensory and motor features (Lines 393-411). We believe this to be unique over system-identification approaches as well. Specifically, our analysis in the comparison with Volterra approaches suggests that it would be very difficult to use system analysis if multiple predictors were involved as this further increases the challenge of obtaining a reasonably fitting model when omitting constraints.

Reviewer #1 (Recommendations for the authors):1) In MINE's first mention (line 43) and in showing that it can capture various response features, the authors seem to imply that MINE is simply the idea of using ANNs to model neural activity as function of either independent or dependent variables. But surely this has already been done many times. I'm having some trouble seeing what precisely is claimed as new in MINE. My thought was that it is also the first and second order interpretation, but that does not come through in the early descriptions of the method.

We agree with the reviewer that it was never properly spelled out what MINE is. We now clarify this point in the introduction (Lines 21-26) where we explicitly state: “Here we introduce “Model-free identification of neural encoding” (MINE). MINE combines convolutional neural networks (CNN) to learn mappings from predictors (stimuli, behavioral actions, internal states) to neural activity (Figure 1) with a deep characterization of this relationship. Using Taylor expansion approaches, MINE reveals the computational complexity such as the nonlinearity of the relationship (Figure 2), characterizes receptive fields as indicators of processing (Figure 3) and reveals the dependence of neural activity on specific predictors or their interactions (Figure 4).

2) Related to this, in L55, the authors remark "Aside from network architecture and initialization conditions MINE is essentially model-free." This is a pretty big constraint, by the model architecture, choice of activation function, etc. STA and STC and systems identification methods more generally seem far more truly model free. This suggests that the framing and title here might need significant reconsideration.

While we agree that the architecture of the model, the type of training performed, etc. are indeed constraints, we stand by our point that MINE is essentially model-free when compared with other approaches in particular in the domain of approximating various functional forms. Specifically, while systems identification approaches can in theory be entirely model-free we demonstrate that in practice (outside a white-noise regime) they really aren’t. We compare effective degrees of freedom of the CNN used by MINE to models that directly fit the Volterra kernels to explicitly address this point (Lines 193-200). We would also like to point out that it is a general theme to reduce the degrees of freedom in systems identification when stimuli are departing from gaussian white noise in order to still be useful. One example is the use of Laguerre basis functions which effectively reduce the degrees of freedom of the fit or constrained regression approaches which we used here in the form of Ridge regression.

3) The authors make several claims about linear models in Figure 1 that I cannot figure out. In particular, at L137, the authors say that the linear model cannot learn the dynamics of the calcium indicator. But why not? If the stimulus is s_t, the kernel is k_t, and the response is r_t = (s*k)_t + eta_t, then this is exactly what a linear model should be able to learn perfectly, directly from the data through a least squares fit at each time point. I don't understand why an expanded linear model should be required to learn the calcium kernel. In 1F, why is the LM not working as well as the ANN for S1 and M2? Those appear to both be simple linear transformations where r_t = (s*k)_t + eta_t: perfect for a linear model and least squares fitting. (In fact, with Gaussian noise, no model should be able to outperform the least squares fit linear model to estimate k, given sufficient data.) Similarly, dS1/dt is still just a linear transformation of S1 and should be easily learned by a linear model, since it's now just r_t = (s*k')_t + eta_t, where k' is the derivative of the calcium kernel. These results – that the authors' linear model is not capturing linear transformations that exactly match the form of the linear model – are truly puzzling.

We are sorry about causing this confusion. Our explicit goal in section one was to compare MINE to a bare-bones linear regression model (no time-shifts, no interactions) which is bound to fail as well as a regression model that is often used in the analysis of calcium imaging data: pre-convolution with a calcium kernel and inclusion of interaction terms but no time-shifted regressors. We now provide citations to works using this type of model and given its prevalence we do believe it to be a useful benchmark. It is however absolutely clear that linear models can learn linear convolutions of the data (such as calcium kernels, computations of the derivative, etc.). We now specifically state that “While other models could clearly be designed to overcome these challenges, this further illustrates the point that a model-based approach is limited to the constraints of the chosen model.” (Line 80), to clarify that we do not claim that MINE does something magic. In the zebrafish section however, we use a linear comparison model that can indeed learn temporal dependencies which is the only reason why it can fit ~40% of the data. As a side note: We initially tried to include a linear model that would include all time-shifts and all interactions in section one, however this model never had any predictive power over validation data after fitting. Again, this could likely have been overcome by carefully fitting a regularization parameter but our main point is that MINE provides a convenient and accessible solution to this problem.

4) One of the big proposed advances of this paper is the interpretability of the fitted ANN model. By doing "Taylor expansion" on the model, the authors pull out linear and quadratic terms that describe the dependency of neurons on different non-neural variables (dependent and independent) in these experiments. I have a few comments on this, some more critical than others. Overall, I was not left with a strong sense of what the benefit is, if any, of this method over standard systems identification approaches.a. This sort of expansion of a functional has more typically been called a Volterra expansion, and there exists a giant literature on estimating Volterra kernels for nonlinear systems. It would be appropriate to cite some of that work and adopt the terminology used in prior work in the field.

We agree with the reviewer and now explicitly state “Since the input to the CNN at the heart of MINE contains information about each predictor across time, the Taylor expansion introduced in the previous section (Figure 2A) is equivalent to the Volterra expansion of a system processing information across time with a filter memory equivalent to the history length of the CNN.” In line with this statement we now also reframe our former STA/STC comparison in terms of Volterra analysis, e.g., stating: “As a comparison, we used regression to directly fit the Volterra kernels (see Methods). Notably, just like MINE, apart from the truncation of the series after the 2nd term, the Volterra analysis is essentially model-free since any function can be approximated using an infinite Volterra series (Volterra, 1959).” We also provide a more comprehensive citation of literature.

b. In comparing their methods to spike-triggered analyses, Figure 4 is not a fair comparison, since it has not decorrelated the stimuli in time, as has been in wide practice in systems identification for quite a while (see Korenberg in the 1990s, Baccus and Meister, 2002). Thus the fair comparison is with S4, and 4 should not be viewed as comparable and quite likely removed.

We agree with the reviewer. Notably, our new approach, directly fitting the Volterra kernels using linear regression circumvents this problem. We have also expanded our comparison to ensure equal footing between MINE and the comparison approach by including regularization of the fit using Ridge regression.

c. In the comparison between the Jacobian/Hessian of the ANN and the STA/STC equivalents in Figures4 and S4, I don't understand the authors' choice of nonlinearities, which are described after line 1105. In particular, g_sta(x) has a first derivative at x=0, but it's second derivative evaluated at x=0 is also non-zero. By adding this to the other function and putting them through a ReLU (equation 19), there will be both linear and quadratic components for both RFa and RFv, even if g_stc is symmetric, simply because the ReLU is approximated by a linear + quadratic (+ higher) terms. This means that with these functions, I would expect the STA to be a linear combination of the two linear projections (RFa and RFv) and I would expect the STC matrix to have eigenvectors spanning the 2-dimensional space of RFa and RFv. It's less clear why this isn't happening in the simulations. RFa could dominate the STA. In Figure 4B, it seems clear that the STC is contaminated by the STA, and given the analysis above, that is not surprising. Is that what is lowering the cosine similarity for the STC analysis in S4? Is only the first eigenvector being considered? If so, it's frequently the case that the STC dimensions also include the STA dimension; often, one reports only the STC components after the STA has been subtracted off. (See for instance Aljadeff et al., 2016.) It looks here as though the authors just report the eigenvector with largest eigenvalue. Is the STC really not working well, or is it just including components of the other filter (appropriately)? It's hard to believe that STC applied to a point nonlinearity acting on two projections of the stimulus would not return eigenvectors that span those two projections of the stimulus.

In the previous comparison we allowed the STA/STC method access to the first two eigenvectors with the largest eigenvalues because of this seeming complication. We would like to note that we did not specifically design the simulation to be as compatible as possible with STA/STC. This would just have repeated a vast body of literature without being practically that relevant. We believe that the important question is whether given a certain system, its filters can be recovered by the analysis at hand. That being said, we now changed our approach to further alleviate this comparison problem: For both MINE and the linear regression fits we combine the first-order vector (J in the case of the CNN, k1 in the case of Volterra) and the second-order matrix (H in the case of the CNN, k2 in the case of Volterra) and extract the principal dynamic modes from this matrix according to Marmarelis, 2004. Here we extract the three eigenvectors with the highest eigenvalues and assign the best-fit cases to the “linear” and “nonlinear” filter (Figure 3). Notably, these filters are always contained within the eigenvectors with the largest two eigenvalues for MINE while the regression based fit sometimes produces a third vector with a larger eigenvalue. Therefore, by allowing to choose from the highest three we in fact give an advantage to the regression based fit. The fit on white-noise input data clearly validates this approach as it flawlessly recovers the filters for both methods. As expected, an ordinarily linear regression model fails in the case of highly correlated stimuli, but filters can be successfully recovered under strongly regularizing conditions, albeit with slightly worse quality than MINE. Furthermore, as regularization increases, the predictive power of the linear model decreases as well. MINE again has a slight edge in this regard.

d. This brings up a conceptual point. The decomposition that the authors perform of the ANN into zeroth, first, second, and higher order terms in Figure 7D could just as easily be written for the raw responses of the neurons. In such a case, with an excellent model, the two expansions should be closely related to one another, and in some cases equivalent. The low-order terms in the Volterra series, when estimated experimentally, depend on higher order terms. (For instance a cubic Volterra term would show up in the linear filter.) Wiener series are typically estimated instead, since these produce an orthogonalized basis, so that this dependence is eliminated (for a given amplitude of inputs X). When computing the STC, one can eliminate the linear component first; that would then be the Wiener estimate of the second order kernel. The second order kernel computed as the Hessian of an ANN is not orthogonalized in the same way, so it might produce different results from STC, but it is not clear that this is what the authors show, or intend to show.

We fully agree with the reviewer and have addressed this point as shown above. Again, as we point out in multiple places of the manuscript, we do not believe MINE to be a magical tool that can do something a carefully crafted model cannot. It simply provides a practical solution to a real-world problem. This is also further illustrated in the greater robustness of MINE to changes in the size of training data compared to directly fitting the Volterra kernels via regression (Figure S3J-I).

e. The authors on L346 say that higher order terms are contained in the ANN, so that the Jacobian will be different from spike triggered covariance analysis. I think this is correct. However, I'm not sure that makes the Hessian or Jacobian in the ANN a better analysis than the systems identification approaches. In my mind, this is potentially the only advantage of doing this ANN method over STA and STC; but if that's the advantage, then I'd want to be shown very clearly an example where this occurs and where the insight into the system was improved by using the Jacobian/Hessian rather than STA/STC. I imagine that such benefits could accrue when the fitted ANN is close to the true set of transformations of the system, but that would reduce the advantages of being able to do this with any old ANN. Overall, these comments get to the point of: under what conditions would the Jacobian/Hessian/ANN method be better or somehow more informative than STA/STC? And why fit to the ANN if one is going to interpret it with methods equivalent to STA/STC that could be performed directly on the data?

The message we intended with L346 was that the nonlinearity, which would be estimated separately for STA/STC, will be contained within the ANN itself which likely influences the retrieved filters. We did not mean to imply anything in relation to the quality of extracting filters and have struck this comment. The possibility of modeling higher-order terms of the CNN model however likely underlies the fact that it better generalizes to validation data (Figure 3C and F). This of course partly results because our nonlinearities are not designed to be well approximated by a quadratic function, i.e. the function that the fit of the Volterra kernels could fully approximate. While one might argue that this makes the comparison “unfair” we would argue that in real-life problems nonlinearities will seldom be constrained to a specific functional form (as we highlight in the introduction). We do believe that our new analysis provides a clear idea of the advantages of MINE, especially when taken in context of the entire manuscript: MINE allows extracting high-quality filters under more conditions than a direct fit of the Volterra kernels, the resulting model has more predictive power and MINE can extract more information from the fits than can easily be obtained by system identification approaches.

Reviewer #2 (Recommendations for the authors):While the paper could certainly be published in the current form, there are some points that as a reader I would like to have seen addressed further. I therefore offer some comments and suggestions that the authors might like to consider.I believe there is a mistake in the formulae given for the Taylor metric in the Methods. Since the R^2 Full is always greater than the R^2 with fewer parameters, the formulae, as written, appear to give a number between 0 and -infinity, whereas I think the intention is that they should give a number between 0 and 1 (as appears in the paper).

We fully agree with the reviewer and apologize for this oversight. We computed the correct quantities in our code but miswrote the formulas in the method section. We have corrected this mistake.

An important consideration when comparing approaches to modeling neuronal responses is how much data is required to fit the model. In the comparison with regression models, or with the spike-triggered average it would be useful to show how the performance of each approach develops with the length of recording.

We agree with the reviewer and have included a corresponding analysis in our discussion of receptive field extraction. Figure S3I-J and Lines 184-192 in the text.

The paper could be improved, and the method made more appealing, by more clearly showing qualitative improvements that their approach offers over variations of linear regression methods. In the zebrafish part, there is a convincing argument made that the method allows more neurons to be fit at some threshold, but direct comparison of the outputs of both approaches is restricted to counting the numbers of ROIs identified in various classes. One possible explanation could be that these differences arise because the model-free method has greater sensitivity for identifying neurons within the same classes that are found with the LM method, but it would be more interesting if the model free method finds particular types of neurons that are clearly missed by the LM method. What could be compelling to see, for example, would be (1) differences in the spatial distributions of neuron classes found with the two methods, e.g. regions where neurons are only found with the model-free method or (2) Evidence that there are specific functional classes that are missed with the LM method – e.g. do the RF clusters vary in how well they are picked up by the LM?

We agree with the reviewer that this is an important point to address. However, whether MINE identifies qualitatively new types or not will also depend on the level of analysis detail. We currently only perform some coarse divisions of neural types to demonstrate various types of information that can be obtained with MINE. As the reviewer suggested we did compare the anatomical distribution of identified types. Making cluster maps (see below where we address our change in visualization) it does indeed appear as though there are parts of the brain where the linear model does not identify stimulus related neurons (dorsal cerebellum, anterior hindbrain). However, we were not comfortable adding this data to the paper. The lack of neurons in these regions could be because the linear model cannot identify “these types of neurons” or simply because it is less sensitive and by chance did not identify neurons within sparse regions.

We did assess how many neurons the linear model identifies in each receptive field cluster and added this information to Figure S7D. While there are no clusters that exclusively contain neurons identified by MINE, there are five clusters where the linear model identifies much fewer neurons than expected based on the overall identified fraction. We discuss this result (Lines 594-601) and point to the fact that this has consequences if the linear model data would be used for clustering in the first place, reducing the number of uniquely identified receptive field clusters from 15 to 4. So while MINE largely increases sensitivity, this has real-world consequences on the analysis which we believe to be important.

It would be helpful to have some more information about the choice of network architecture, and how this was optimized. For example, 80 features in the convolutional layer seems like a lot for these relatively simple data sets, and it would be good to understand better how this number was chosen and how this choice can affect the outcome.

We did not systematically explore network architectures and did not optimize the architecture. In part because we wanted to avoid pre-conceiving an architecture that we thought might be best for the data at hand. We have now clarified this in the methods section “Design and training of convolutional network.” In that section we also provide more detail on our rationale for using 80 convolutional layers. We do not think that 80 features are in fact extracted, but having a large number of layers means a large variation in random features that are extracted pre-training which can aid in network training. Since our convolutional layers are linear, each node in the first dense layer in fact uses one weighted average filter that acts on the input data.

Intuitively, it seems like the temporal effects of the calcium indicator response would ideally be applied downstream of the computation of the neural response, but, in the text, this step is described as happening in the first layer of linear convolutional filters. Is this a constraint of the network design, and are there any practical consequences for the possible computations captured by the model?

We agree with the reviewer that in measurements of brain activity, the convolution by the calcium indicator occurs after any other transformations. We perform all convolutions at the input layer to increase the simplicity of the network. After the initial convolutional layers all timing information is lost in the network which reduces the number of required parameters over a design that allows for convolution at the input and just before the output node. While this does limit expressivity of the network somewhat (since we do not allow for convolutions of the results of the nonlinear operations as would be expected in a calcium imaging experiment) we think that these effects are small and justified by the gain in simplicity in the network. Since we fit separate networks for each neuron, having fewer parameters leads to substantial speed gains. We note however, that all procedures implemented in MINE are agnostic to network architecture and it wouldn’t be a fundamental problem to make this change.

I don't like that the distributions of neuronal types seem to be shown in the form of renderings of anatomical masks highlighting regions of enrichment. This rather obscures the true distributions, and makes them appear more cleanly separated than they are. For me, these visualizations don't offer much information that goes beyond the tabulation of regions. A direct representation of the density of different functional classes, as is often used in the field, would be richer in information, and might be a way to highlight qualitative differences between the populations identified using the LM and model-free approaches. Even if it is the case that there are not strong differences in the distributions of functional types, this by itself would be an interesting observation that merits discussion.

We agree with the reviewer and have adjusted our visualizations. While we keep the bar-plot style tabulations for the comparison of stimulus/behavior/mixed and swim start/vigor/direction we now plot centroid maps in all cases. We created these maps using spatial clustering. In brief, we merge neurons into spatial clusters based on proximity (threshold of 5um) and then display all clusters that have at least 10 neurons. While this does not display the entire population of neurons it does give a better sense of the overall distribution. Structure is also visible when showing all neurons, however in 2D projections differences between areas where neurons actually group together versus areas in which there are multiple single neurons across 3D space are obscured. We therefore opted for this middle-ground between thresholding based on anatomical regions and a plot with disregard for spatial grouping. We hope that this addresses the point raised by the reviewer.

I would like to see more discussion of how the neural network models could be explored or leverage to give more insight into the responses of individual neurons. For example, having identified the relevant variables, and model complexity, could this information be used to explore simpler models in a more constrained way? Could visualization approaches like those used in Heras et al. 2019 https://doi.org/10.1371/journal.pcbi.1007354, be used to explore how the network is combining task variables?

We agree with the reviewer and now exploit a similar visualization for the mixed selectivity neurons (Figure 7D and S7E). Since our inputs are stimuli/behaviors across time we cannot directly visualize the effect of one variable on the output. We therefore exploit the receptive fields to provide varying strengths of inputs along the different axis which we refer to as drive. We think that this is also another good example for the power of MINE, exploiting the ability of extracting receptive fields together with the presence of a predictive model that encapsulates nonlinear relationships.

Since the function is scalar valued, I believe that what is referred to in the paper as the Jacobian is also simply the gradient of the function. Perhaps this could be mentioned explicitly to aid understanding for a broad audience.

We fully agree with the reviewer and have adjusted our writing. We still refer to the first order derivatives as J in the formulas but refer to it as the gradient of the function (e.g., Lines 101-111 in the section about computational complexity).

Overall, I congratulate the authors on developing this useful method, and I look forward to seeing it published.

We thank the reviewer for their assessment and hope that our additions have improved the paper.

Reviewer #3 (Recommendations for the authors):1. Overall, as mentioned in my Public Review section, I believe the paper is very valuable and merits publication in eLife. I do think that it would be a pity if the manuscript was not carefully re-read and rewritten to make it more accessible for the average reader who may want to implement this analysis on their data. An option would be to shift more material to the Methods and use the main text to convey the method and compare it with the available ones for the 3 datasets that are treated.

We greatly appreciate the reviewer’s positive assessment of our work. We hope that our rewrite addresses concerns with respect to readability and accessibility. We think that readers will now have an easier time following along, after we rearranged the sections of the paper and simplified our analysis of non-linearity. We also attempted to shift more details to the methods, however this wasn’t entirely possible in all places. We had considered presenting MINE as a black-box-model essentially demonstrating its usefulness on the three types of data we present (ground-truth, cortex and zebrafish). We would then have split details to some form of appendix as the reviewer suggests. However, ultimately we felt that this is also not a satisfying solution. While it certainly increases accessibility we worried that it would make it harder to judge the validity of our approach. We hope that we found a satisfying middle-ground in the revision.

2. It would help to understand the various thresholds that appear throughout. I am, for example, not sure what to conclude from the sentence that starts on line 224 or how to interpret the red dashed line in Figure 5B (line 421).

We agree with the reviewer that it is important to clarify thresholds. We have removed the threshold from the Taylor decomposition section (now around line 226) and corresponding figures as it is indeed not very useful. What we tried to say is that it is possible to identify thresholds which can separate true contributing predictors from noise. However, since we do not use this threshold later it really doesn’t serve a purpose.

We have clarified our choice of threshold for the cortical dataset (now around lines 248-252) which is much lower than the zebrafish one, since the data does not have cellular resolution. It is therefore very likely that every component we fit contains activity from many neurons that aren’t task related, limiting the amount of variance that could possibly be explained by a model.

We also added an explanation for our threshold used on the zebrafish data and why it is considerably higher than that used on cortical data (Line 319).

3. Given that the zebrafish experiments were performed by the authors, could they comment on the reliability (across animals) in finding the functional activity described in Figures 6 and 7?

We thank the reviewer for this question and we have added this important information to the paper. Specifically, in Figure 6E we now state for each functional type in how many fish it was identified. All but four types were identified across at least 10 fish, with the lowest number being four fish. Accordingly, we state in the text that each type was identified in multiple fish (Line 337). We also counted how many fish contribute to each receptive field cluster. We report in the text that each type was identified in at least 15 fish (Line 383).

4. The code and data availability is exemplary.

We thank the reviewer for this assessment.

[Editors' note: further revisions were suggested prior to acceptance, as described below.]

The manuscript has been much improved, but there is one large and a few small remaining issues that need to be addressed, as outlined below:There is only one issue that remains, and it is the issue of calling this a "model-free" approach, noted by R1. Since no analysis is truly model free, there is a spectrum of how much constraint a model puts on an analysis. There are other approaches discussed in the paper that are closer to the model free end than this one (like STA, STC, MID), so this term does not seem like the right one to use in this case. The authors should find a more accurate description of what their method brings to field.

We thank the reviewers for again engaging with our manuscript and providing constructive criticism. We have made textual changes (and one figure clarification suggested by R1) to address the remaining concerns. This specifically includes a change to the title and moniker of the method to refrain calling it “model-free”. We agree with R1 that it is important to be precise about this point, and now focus on the idea that the neural network allows the method to be flexible in terms of the function/model it can implement. This was our reason for calling it “model-free” in the first place – not that it is free from constraints on *parameters* but that it is free(er) from constraints on the *functional form*. We therefore changed the title to “Model discovery to link neural activity to behavioral tasks”. We kept the acronym “MINE” and simply struck the term “free”, referring to it as “Model identification of neural encoding” throughout the text. We only kept one reference to the “model-free” idea, when we talk about the lack of assumptions on the underlying distribution of data but added the caveat “largely model-free” in this statement. All other references to the method being “model-free” have been struck from the text.

Reviewer #1 (Recommendations for the authors):I appreciate the care that the authors have taken in addressing the concerns raised in the first round. I believe that the major hurdles were cleared in the revision. However, I do have a few comments that I believe should be addressed before final acceptance/publication.

We thank the reviewer for their assessment and believe that our new changes have addressed the remaining concerns.

1) This method is definitely not "model free". It is a relaxed model requirement, or potentially a very general model, or a model with few assumptions, but I don't think it's just a matter of opinion as to whether it's "model-free". Especially in comparison with STA and STC and Maximally informative dimensions, which all have even less of an underlying model framework… At a very basic level, the title and acronym must be accurate. Later in the paper, the authors argue that this method's model is actually providing important regularizing power in computing STC, etc. So it is definitely not model free, and the authors must find a different description for the title (and the acronym). I'm sure they can do this, but some suggestions: "Flexible INE"? CNNINE? "Canine"? "Convolutional neurAl Network Identification of Neural Encoding"? Something else? "Model-agnostic"?

We have changed the title of the manuscript as well as the text to be precise about this point. We now emphasize the idea that the advantage of MINE is that it is flexible with respect to the functional form it can implement, turning model-based analysis on its head: Instead of defining a model and subsequently fitting parameters, MINE will discover a model which we subsequently characterize. As stated above, we therefore changed the title to “Model discovery to link neural activity to behavioral tasks”. We kept the acronym “MINE” and simply struck the term “free”, referring to it as “Model identification of neural encoding” throughout the text. We only kept one reference to the “model-free” idea, when we talk about the lack of assumptions on the underlying distribution of data but added the caveat “largely model-free” in this statement. All other references to the method being “model-free” have been struck from the text. The changes made to address this point are highlighted in the tracked version of the manuscript at the end of this reviewer response.

2) Figure 7BC. The temporal kernels found by this method are a bit weird and seem not very biological – much of the power lies at 0 and 10 seconds, the two ends of the filter… Is this because the filters aren't long enough? These look quite different from true filters I'd expect to find in neurons, both in their duration and shape. I think this deserves some kind of explanation. I apologize that I didn't note this on the first round – I clearly got distracted by some other parts of the analysis.

We agree with the reviewer that this should be addressed. To this end, we have added a caveat to the result section, where we now explicitly state:

“We note that some of the uncovered receptive fields (e.g., clusters 1 and 4) have the largest departures from 0 at the starts and ends of the receptive fields. This might indicate that the chosen history length (here 10 s) is too short and does not cover the entire timescale of processing which could be a result of the rather slow nuclear calcium indicator we chose for this study.”